# Projected GANs Converge Faster

**Axel Sauer**[1,2]    **Kashyap Chitta**[1,2]    **Jens Müller**[3]    **Andreas Geiger**[1,2]

[1]University of Tübingen    [2]Max Planck Institute for Intelligent Systems, Tübingen
[3]Computer Vision and Learning Lab, University Heidelberg
[2]{firstname.lastname}@tue.mpg.de    [3]{firstname.lastname}@iwr.uni-heidelberg.de

## Abstract

Generative Adversarial Networks (GANs) produce high-quality images but are challenging to train. They need careful regularization, vast amounts of compute, and expensive hyper-parameter sweeps. We make significant headway on these issues by projecting generated and real samples into a fixed, pretrained feature space. Motivated by the finding that the discriminator cannot fully exploit features from deeper layers of the pretrained model, we propose a more effective strategy that mixes features across channels and resolutions. Our Projected GAN improves image quality, sample efficiency, and convergence speed. It is further compatible with resolutions of up to one Megapixel and advances the state-of-the-art Fréchet Inception Distance (FID) on twenty-two benchmark datasets. Importantly, Projected GANs match the previously lowest FIDs up to 40 times faster, cutting the wall-clock time from 5 days to less than 3 hours given the same computational resources.

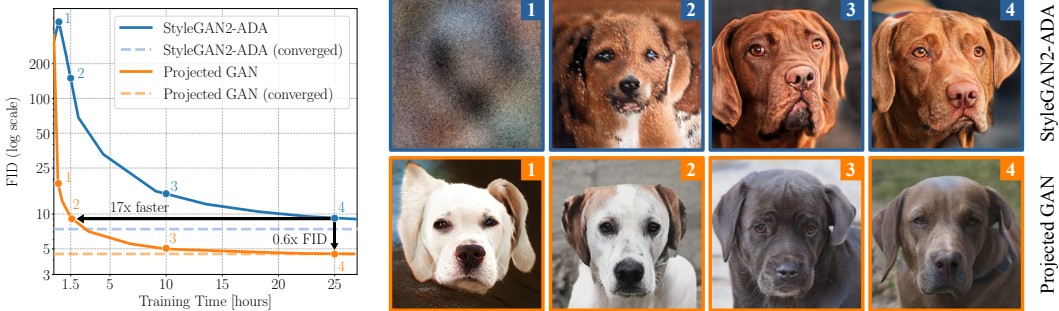

Figure 1: **Convergence with Projected GANs.** Evolution of samples for a fixed latent code during training on the AFHQ-Dog dataset [5]. We find that discriminating features in the projected feature space speeds up convergence and yields lower FIDs. This finding is consistent across many datasets.

## 1    Introduction

A Generative Adversarial Network (GAN) consists of a generator and a discriminator. For image synthesis, the generator's task is to generate an RGB image; the discriminator aims to distinguish real from fake samples. On closer inspection, the discriminator's task is two-fold: First, it projects the real and fake samples into a meaningful space, i.e., it learns a representation of the input space. Second, it discriminates based on this representation. Unfortunately, training the discriminator jointly with the generator is a notoriously hard task. While discriminator regularization techniques help to balance the adversarial game [31], standard regularization methods like gradient penalties [36] are susceptible to hyperparameter choices [26] and can lead to a substantial decrease in performance [4].

In this paper, we explore the utility of pretrained representations to improve and stabilize GAN training. Using pretrained representations has become ubiquitous in computer vision [29, 30, 48]

35th Conference on Neural Information Processing Systems (NeurIPS 2021).

and natural language processing [18, 45, 47]. While combining pretrained perceptual networks [58] with GANs for image-to-image translation has led to impressive results [14, 49, 59, 64], this idea has not yet materialized for unconditional noise-to-image synthesis. Indeed, we confirm that a naïve application of this idea does not lead to state-of-the-art results (Section 4) as strong pretrained features enable the discriminator to dominate the two-player game, resulting in vanishing gradients for the generator [2]. In this work, we demonstrate how these challenges can be overcome and identify two key components for exploiting the full potential of pretrained perceptual feature spaces for GAN training: **feature pyramids** to enable multi-scale feedback with multiple discriminators and **random projections** to better utilize deeper layers of the pretrained network.

We conduct extensive experiments on small and large datasets with a resolution of up to $1024^2$ pixels. Across all datasets, we demonstrate state-of-the-art image synthesis results at significantly reduced training time (Fig. 1). We also find that Projected GANs increase data efficiency and avoid the need for additional regularization, rendering expensive hyperparameter sweeps unnecessary. Code, models, and supplementary videos can be found on the project page https://sites.google.com/view/projected-gan.

## 2 Related Work

We categorize related work into two main areas: pretraining for GANs and discriminator design.

**Pretrained Models for GAN Training.** Work on leveraging pretrained representations for GANs can be divided into two categories: First, transferring parts of a GAN to a new dataset [15, 38, 65, 71] and, second, using pretrained models to control and improve GANs. The latter is advantageous as pretraining does not need to be adversarial. Our work falls into this second category. Pretrained models can be used as a guiding mechanism to disentangle causal generative factors [54], for text-driven image manipulation [44], matching the generator activations to inverted classifiers [19, 56], or to generate images via gradient ascent in the latent space of a generator [41]. The non-adversarial approach of [53] learns generative models with moment matching in pretrained models; however, the results remain far from competitive to standard GANs. An established method is the combination of adversarial and perceptual losses [21]. Commonly, the losses are combined additively [10, 14, 32, 52, 64]. Additive combination, however, is only possible if a reconstruction target is available, e.g., in paired image-to-image translation settings [74]. Instead of providing the pretrained network with a reconstruction target, Sungatullina et al. [59] propose to optimize an adversarial loss on frozen VGG features [58]. They show that their approach improves CycleGAN [74] on image translation tasks. In a similar vein, [49] recently proposed a different perceptual discriminator. They utilize a pretrained VGG and connect its features with the prediction of a pretrained segmentation network. The combined features are fed into multiple discriminators at different scales. The two last approaches are specific to the image-to-image translation task. We demonstrate that these methods do not work well for the more challenging unconditional setting where the entire image content is synthesized from a random latent code.

**Discriminator Design.** Much work on GANs focuses on novel generator architectures [4, 26, 27, 69], while the discriminator often remains close to a vanilla convolutional neural network or mirrors the generator. Notable exceptions are [55, 70] which utilize an encoder-decoder discriminator architecture. However, in contrast to us, they neither use pretrained features nor random projections. A different line of work considers a setup with multiple discriminators, applied to either the generated RGB image [8, 13] or low-dimensional projections thereof [1, 40]. The use of several discriminators promises improved sample diversity, training speed, and training stability. However, these approaches are not utilized in current state-of-the-art systems because of diminishing returns compared to the increased computational effort. Providing multi-scale feedback with one or multiple discriminators has been helpful for both image synthesis [23, 24] and image-to-image translation [43, 64]. While these works interpolate the RGB image at different resolutions, our findings indicate the importance of multi-scale *feature maps*, showing parallels to the success of pyramid networks for object detection [34]. Lastly, to prevent overfitting of the discriminator, differentiable augmentation methods have recently been proposed [25, 63, 72, 73]. We find that adopting these strategies helps exploit the full potential of pretrained representations for GAN training.

# 3 Projected GANs

GANs aim to model the distribution of a given training dataset. A generator $G$ maps latent vectors $\mathbf{z}$ sampled from a simple distribution $\mathbb{P}_{\mathbf{z}}$ (typically a normal distribution) to corresponding generated samples $G(\mathbf{z})$. The discriminator $D$ then aims to distinguish real samples $\mathbf{x} \sim \mathbb{P}_{\mathbf{x}}$ from the generated samples $G(\mathbf{z}) \sim \mathbb{P}_{G(\mathbf{z})}$. This basic idea results in the following minimax objective

$$\min_G \max_D \left( \mathbb{E}_{\mathbf{x}}[\log D(\mathbf{x})] + \mathbb{E}_{\mathbf{z}}[\log(1 - D(G(\mathbf{z})))] \right) \tag{1}$$

We introduce a set of feature projectors $\{P_l\}$ which map real and generated images to the discriminator's input space. Projected GAN training can thus be formulated as follows

$$\min_G \max_{\{D_l\}} \sum_{l \in \mathcal{L}} \left( \mathbb{E}_{\mathbf{x}}[\log D_l(P_l(\mathbf{x}))] + \mathbb{E}_{\mathbf{z}}[\log(1 - D_l(P_l(G(\mathbf{z}))))] \right) \tag{2}$$

where $\{D_l\}$ is a set of independent discriminators operating on different feature projections. Note that we keep $\{P_l\}$ fixed in (2) and only optimize the parameters of $G$ and $\{D_l\}$. The feature projectors $\{P_l\}$ should satisfy two necessary conditions: they should be differentiable and provide sufficient statistics of their inputs, i.e., they should preserve important information. Moreover, we aim to find feature projectors $\{P_l\}$ which turn the (difficult to optimize) objective in (1) into an objective more amenable to gradient-based optimization. We now show that a projected GAN indeed matches the distribution in the projected feature space, before specifying the details of our feature projectors.

## 3.1 Consistency

The projected GAN objective in (2) no longer optimizes directly to match the true distribution $\mathbb{P}_T$. To understand the training properties under ideal conditions, we consider a more generalized form of the consistency theorem of [40]:

**Theorem 1.** *Let $\mathbb{P}_T$ denote the density of the true data distribution and $\mathbb{P}_G$ the density of the distribution the Generator $G$ produces. Let $P_l \circ T$ and $P_l \circ G$ be the functional composition of the differentiable and fixed function $P_l$ and the true/generated data distribution, and $\mathbf{y}$ be the transformed input to the discriminator. For a fixed $G$, the optimal discriminators are given by*

$$D_{l,G}^*(\mathbf{y}) = \frac{\mathbb{P}_{P_l \circ T}(\mathbf{y})}{\mathbb{P}_{P_l \circ T}(\mathbf{y}) + \mathbb{P}_{P_l \circ G}(\mathbf{y})}$$

*for all $l \in \mathcal{L}$. In this case, the optimal $G$ under (2) is achieved iff $\mathbb{P}_{P_l \circ T} = \mathbb{P}_{P_l \circ G}$ for all $l \in \mathcal{L}$.*

A proof of the theorem is provided in the appendix. From the theorem, we conclude that a feature projector $P_l$ with its associated discriminator $D_l$ encourages the generator to match the true distribution along the marginal through $P_l$. Therefore, at convergence, $G$ matches the generated and true distributions in feature space. The theorem also holds when using stochastic data augmentations [25] before the deterministic projections $P_l$.

## 3.2 Model Overview

Projecting to and training in pretrained feature spaces opens up a realm of new questions which we address below. This section will provide an overview of the general system and is followed by extensive ablations of each design choice. As our feature projections affect the discriminator, we focus on $P_l$ and $D_l$ in this section and postpone the discussion of generator architectures to Section 5.

**Multi-Scale Discriminators.** We obtain features from four layers $L_l$ of a pretrained feature network $F$ at resolutions ($L_1 = 64^2, L_2 = 32^2, L_3 = 16^2, L_4 = 8^2$). We associate a separate discriminator $D_l$ with the features at layer $L_l$, respectively. Each discriminator $D_l$ uses a simple convolutional architecture with spectral normalization [37] at each convolutional layer. We observe better performance if all discriminators output logits at the same resolution ($4^2$). Accordingly, we use fewer down-sampling blocks for lower resolution inputs. Following common practice, we sum all logits for computing the overall loss. For the generator pass, we sum the losses of all discriminators. More complex strategies [1, 13] did not improve performance in our experiments.

**Random Projections.** We observe that features at deeper layers are significantly harder to cover, as evidenced by our experiments in Section 4. We hypothesize that a discriminator can focus on a subset of the feature space while wholly disregarding other parts. This problem might be especially

prominent in the deeper, more semantic layers. Therefore, we propose two different strategies to dilute prominent features, encouraging the discriminator to utilize all available information equally. Common to both strategies is that they mix features using differentiable random projections which are fixed, i.e., after random initialization, the parameters of these layers are not trained.

*Cross-Channel Mixing (CCM).* Empirically, we found two properties to be desirable: (i) the random projection should be information preserving to leverage the full representational power of $F$, and (ii) it should not be trivially invertible. The easiest way to mix across channels is a $1 \times 1$ convolution. A $1 \times 1$ convolution with an equal number of output and input channels is a generalization of a permutation [28] and consequently preserves information about its input. In practice, we find that more output channels lead to better performance as the mapping remains injective and therefore information preserving. Kingma et al. [28] initialize their convolutional layers as a random rotation matrix as a good starting point for optimization. We do not find this to improve GAN performance (see Appendix), arguably since it violates (ii). We therefore randomly initialize the weights of the convolutional layer via Kaiming initialization [16]. Note that we do not add any activation functions. We apply this random projection at each of the four scales and feed the transformed feature to the discriminator as depicted in Fig. 2.

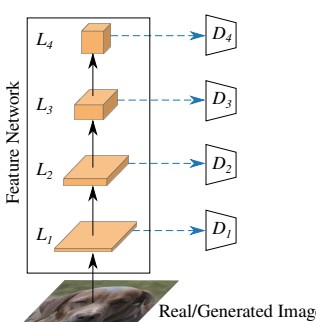

Figure 2: **CCM** (dashed blue arrows) employs $1 \times 1$ convolutions with random weights.

*Cross-Scale Mixing (CSM).* To encourage feature mixing *across* scales, CSM extends CCM with random $3 \times 3$ convolutions and bilinear upsampling, yielding a U-Net [50] architecture, see Fig. 3. However, our CSM block is simpler than a vanilla U-Net [50]: we only use a single convolutional layer at each scale. As for CCM, we utilize Kaiming initialization for all weights.

**Pretrained Feature Networks.** We ablate over varying feature networks. First, we investigate different versions of EfficientNets, which allow for direct control over model size versus performance. EfficientNets are image classification models trained on ImageNet [7] and designed to provide favorable accuracy-compute tradeoffs. Second, we use ResNets of varying sizes. To analyze the dependency on ImageNet features (Section 4.3), we also consider R50-CLIP [46], a ResNet optimized with a contrastive language-image objective on a

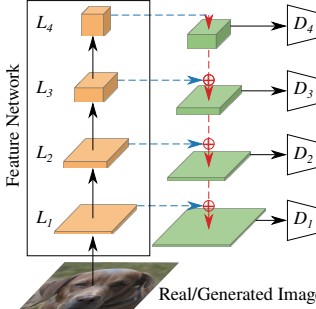

Figure 3: **CSM** (dashed red arrows) adds random $3 \times 3$ convolutions and bilinear upsampling, yielding a U-Network.

dataset of 400 million (image, text) pairs. Lastly, we utilize a vision transformer architecture (ViT-Base) [9] and its efficient follow-up (DeiT-small distilled) [62]. We do not choose an inception network [60] to avoid strong correlations with the evaluation metric FID [17]. In the appendix, we also evaluate several other neural and non-neural metrics to rule out correlations. These additional metrics reflect the rankings obtained by FID.

In the following, we conduct a systematic ablation study to analyze the importance and best configuration of each component in our Projected GAN model, before comparing it to the state-of-the-art.

# 4   Ablation Study

To determine the best configuration of discriminators, mixing strategy, and pretrained feature network, we conduct experiments on LSUN-Church [67], which is medium-sized (126k images) and reasonably visually complex, using a resolution of $256^2$ pixels. For the generator $G$ we use the generator architecture of FastGAN [35], consisting of several upsampling blocks, with additional skip-layer-excitation blocks. Using a hinge loss [33], we train with a batch size of 64 until 1 million real images have been shown to the discriminator, a sufficient amount for $G$ to reach values close to convergence. If not specified otherwise, we use an EfficientNet-Lite1 [61] feature network in this section. We found that discriminator augmentation [25, 63, 72, 73] consistently improves the performance of all methods, and is required to reach state-of-the-art performance. We leverage differentiable data-augmentation [72] which we found to yield the best results in combination with the FastGAN generator.

| Discriminator(s) | $rel\text{-}FD_1\downarrow$ | $rel\text{-}FD_2\downarrow$ | $rel\text{-}FD_3\downarrow$ | $rel\text{-}FD_4\downarrow$ | $rel-FID\downarrow$ |
|---|---|---|---|---|---|
| **No Projection** | | | | | |
| on $L_1$ | 0.56 | 0.32 | 0.31 | 0.55 | 0.66 |
| on $L_1, L_2$ | **0.35** | **0.21** | **0.23** | **0.47** | **0.53** |
| on $L_1, L_2, L_3$ | 0.42 | 0.26 | 0.28 | 0.64 | 0.90 |
| on $L_1, L_2, L_3, L_4$ | 0.46 | 0.34 | 0.38 | 0.79 | 1.15 |
| on $L_2, L_3, L_4$ | 0.95 | 0.67 | 0.71 | 1.19 | 1.99 |
| on $L_3, L_4$ | 2.14 | 1.41 | 1.18 | 1.99 | 3.46 |
| on $L_4$ | 10.92 | 5.74 | 2.56 | 2.79 | 5.08 |
| Perceptual $D$ | 2.98 | 1.76 | 1.20 | 1.89 | 2.73 |
| **CCM** | | | | | |
| on $L_1$ | **0.27** | 0.21 | 0.26 | 0.50 | 0.59 |
| on $L_1, L_2$ | **0.27** | **0.18** | **0.21** | **0.41** | **0.48** |
| on $L_1, L_2, L_3$ | 0.31 | 0.25 | 0.24 | 0.54 | 0.67 |
| on $L_1, L_2, L_3, L_4$ | 0.53 | 0.34 | 0.34 | 0.59 | 0.77 |
| Perceptual $D$ | 5.33 | 3.06 | 2.14 | 1.09 | 4.77 |
| **CCM + CSM** | | | | | |
| on $L_1$ | 0.34 | 0.25 | 0.19 | 0.35 | 0.44 |
| on $L_1, L_2$ | **0.21** | 0.18 | 0.16 | 0.27 | 0.31 |
| on $L_1, L_2, L_3$ | 0.41 | 0.26 | 0.17 | 0.23 | 0.29 |
| on $L_1, L_2, L_3, L_4$ | 0.26 | **0.16** | **0.13** | **0.16** | **0.24** |
| Perceptual $D$ | 2.53 | 1.37 | 0.89 | 0.43 | 2.13 |

Table 1: **Feature Space Fréchet Distances.** We aim to find the best combination of discriminators and random projections to fit the distributions in feature network $F$. We show the relative FD at different layers of $F$ ($rel\text{-}FD_i$) between 50k generated and real images on LSUN-Church. $rel\text{-}FD_i$ is **normalized** using the baseline Fréchet Distances for a model with a standard single RGB image discriminator. Hence, values $> 1$ indicate worse performance than the RGB baseline. We report $rel\text{-}FD$ for four layers of an EfficientNet ($L_1, L_2, L_3$ and $L_4$ from shallow to deep), as well as relative Fréchet Inception Distance (FID) [17]. Note that $rel\text{-}FD_i$ should not be compared between different feature spaces, i.e., only within-column comparisons are meaningful. Blue boxes highlight the layers which we supervise via independent discriminators. The green box corresponds to a perceptual discriminator [59], which takes in all feature maps at once.

## 4.1 Which feature network layers are most informative?

We first investigate the relevance of independent multi-scale discriminators. For this experiment, we do not use feature mixing. To measure how well $G$ fits a particular feature space, we employ the Fréchet Distance (FD) [12] on the spatially pooled features denoted as $FD_i$ for layer $i$. FDs across different feature spaces are not directly comparable. Therefore, we train a GAN baseline with a standard RGB discriminator, record $FD_i^{RGB}$ at each layer and quantify the relative improvement via the fraction $rel\text{-}FD_i = FD_i/FD_i^{RGB}$. We also investigate a perceptual discriminator [59], where feature maps are fed into different layers of the *same* discriminator to predict a single logit.

The results in Table 1 (No Projection) show that two discriminators are better than one and improve over the vanilla RGB baseline. Surprisingly, adding discriminators at deep layers hurts performance. We conclude that these more semantic features do not respond well to direct adversarial losses. We also experimented with discriminators at resized versions of the original image, but could not find a setting of hyperparameters and architectures that improves over the single image baseline. Omitting the discriminators on the shallow features decreases performance, which is anticipated, as these layers contain most of the information about the original image. A similar effect has been observed for feature inversion [11] – the deeper the layer, the harder it is to reconstruct its input. Lastly, we observe that independent discriminators outperform the perceptual discriminator by a significant margin.

| | EfficientNet | | | | | ResNet | | | Transformer | |
|---|---|---|---|---|---|---|---|---|---|---|
| | lite0 | lite1 | lite2 | lite3 | lite4 | R18 | R50 | R50-CLIP | DeiT | ViT |
| Params (M) $\downarrow$ | 2.96 | 3.72 | 4.36 | 6.42 | 11.15 | 11.18 | 23.51 | 23.53 | 92.36 | 317.52 |
| IN top-1 $\uparrow$ | 75.48 | 76.64 | 77.47 | 79.82 | 81.54 | 69.75 | 79.04 | N/A | 85.42 | 85.16 |
| FID $\downarrow$ | 2.53 | 1.65 | 1.69 | 1.79 | 2.35 | 4.16 | 4.40 | 3.80 | 2.46 | 12.38 |

Table 2: **Pretrained Feature Networks Study**. We train the projected GAN with different pretrained feature networks. We find that compact EfficientNets outperform both ResNets and Transformers.

### 4.2 How can we best utilize the pretrained features?

Given the insights from the previous section, we aim to improve the utilization of deep features. For this experiment, we only investigate configurations that include discriminators at high resolutions. Table 1 (CCM and CCM + CSM) presents the results for both mixing strategies. CCM moderately decreases the FDs across all settings, confirming our hypothesis that mixing channels results in better feedback for the generator. When adding CSM, we achieve another notable improvement across all configurations. Especially $rel\text{-}FD_i$ at deeper layers are significantly decreased, demonstrating CSM's usefulness to leverage deep semantic features. Interestingly, we observe that the best performance is now obtained by combining all four discriminators. A perceptual discriminator is again inferior to multiple discriminators. We remark that integrating the original image, via an independent discriminator or CCM or CSM always resulted in worse performance. This failure suggests that naïvely combining non-projected with projected adversarial optimization impairs training dynamics.

### 4.3 Which feature network architecture is most effective?

Using the best setting determined by the experiments above (CCM + CSM with four discriminators), we study the effectiveness of various perceptual feature network architectures for Projected GAN training. To ensure convergence, also for larger architectures, we train for 10 million images. Table 2 reports the FIDs achieved on LSUN-Church. Surprisingly, we find that there is no correlation with ImageNet accuracy. On the contrary, we observe lower FIDs for smaller models (e.g., EfficientNets-lite). This observation indicates that a more *compact* representation is beneficial while at the same time reducing computational overhead and consequently training time. R50-CLIP slightly outperforms its R50 counterpart, indicating that ImageNet features are not required to achieve low FID. For the sake of completeness, we also train with randomly initialized feature networks, which, however, converge to much higher FID values (see Appendix). In the following, we thus use EfficientNet-Lite1 as our feature network.

## 5 Comparison to State-of-the-Art

This section conducts a comprehensive analysis demonstrating the advantages of Projected GANs with respect to state-of-the-art models. Our experiments are structured into three sections: evaluation of convergence speed and data efficiency (5.1), and comparisons on large (5.2) and small (5.3) benchmark datasets. We cover a wide variety of datasets in terms of size (hundreds to millions of samples), resolution ($256^2$ to $1024^2$), and visual complexity (clip-art, paintings, and photographs).

**Evaluation Protocol.** We measure image quality using the Fréchet Inception Distance (FID) [17]. Following [26, 27], we report the FID between 50k generated and all real images. We select the snapshot with the best FID for each method. In addition to image quality, we include a metric to evaluate convergence. As in [25], we measure training progress based on the number of real images shown to the discriminator (*Imgs*). We report the number of images required by the model for the FID to reach values within 5% of the best FID over training. In the appendix, we also report other metrics that are less benchmarked in GAN literature: KID [3], SwAV-FID [39], precision and recall [51]. Unless otherwise specified, we follow the evaluation protocol of [20] to facilitate fair comparisons. Specifically, we compare all approaches given the same fixed number of images (10 million). With this setting, each experiment takes roughly 100-200 GPU hours on a NVIDIA V100, for more details we refer to the appendix.

**Baselines.** We use StyleGAN2-ADA [25] and FastGAN [35] as baselines. StyleGAN2-ADA is the strongest model on most datasets in terms of sample quality, whereas FastGAN excels in training speed. We implement these baselines and our Projected GANs within the codebase provided by the authors of StyleGAN2-ADA [25]. For each model, we ran two kinds of data augmentation:

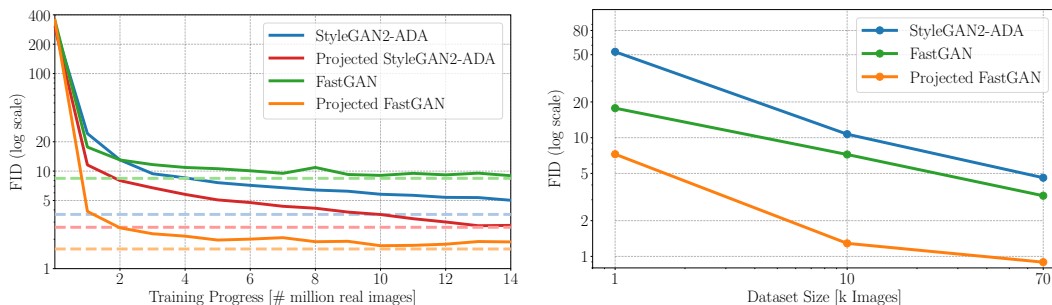

Figure 4: **Training Properties.** Left: Projected FastGAN surpasses the best FID of StyleGAN2 (at 88 M images) after just 1.1 M images on LSUN-Church. Right: Projected FastGAN yields significantly improved FID scores, even when using subsets of CLEVR with 1k and 10k samples.

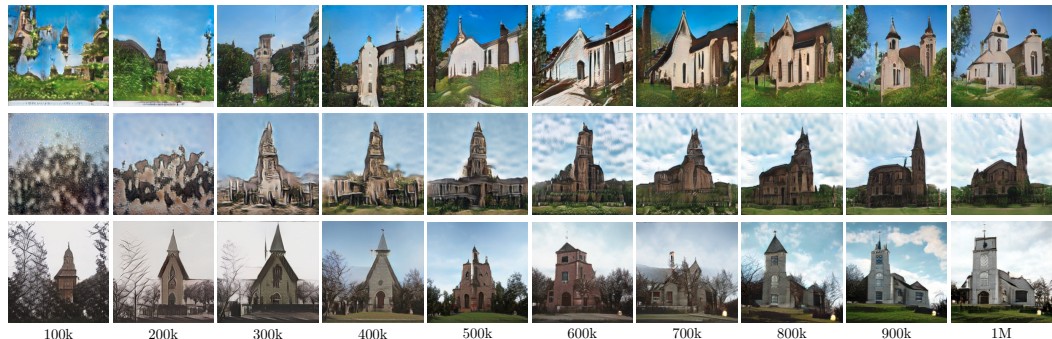

Figure 5: **Training progress on LSUN church at** $256^2$ **pixels.** Shown are samples for a fixed noise vector **z** over k images. From top to bottom: FastGAN, StyleGAN2-ADA, Projected GAN.

differentiable data-augmentation [72] and adaptive discriminator augmentation [25]. We select the better performing augmentation strategy per model. For all baselines and datasets, we perform data amplification through x-flips. Projected GANs use the same generator and discriminator architecture and training hyperparameters (learning rate and batch size) for all experiments. For high-resolution image generation, additional upsampling blocks are included in the generator to match the desired output resolution. We carefully tune all hyper-parameters for both baselines for best results: we find that FastGAN is sensitive to the choice of batch size, and StyleGAN2-ADA to the learning rate and R1 penalty. The appendix documents additional implementation details used in each of our experiments.

### 5.1 Convergence Speed and Data Efficiency

Following [20] and [68], we analyze the training properties of Projected GANs on LSUN-Church at an image resolution of $256^2$ pixels and on the 70k CLEVR dataset [22]. In this section, we also train longer than 10 M images if necessary, as we are interested in convergence properties.

**Convergence Speed.** We apply projected GAN training for both the style-based generator of Style-GAN2 and the standard generator with a single input noise vector of FastGAN. As shown in Fig. 4 (left), FastGAN converges quickly but saturates at a high FID. StyleGAN2 converges more slowly (88 M images) but reaches a lower FID. Projected GAN training improves both generators. Particularly for FastGAN, improvements in both convergence speed and final FID are significant while improvements for StyleGAN2 are less pronounced. Remarkably, *Projected FastGAN* reaches the previously best FID of StyleGAN2 after experiencing only 1.1 M images as compared to 88 M of StyleGAN2. In wall clock time, this corresponds to less than 3 hours instead of 5 days. Hence, from now on, we utilize the FastGAN generator and refer to this model simply as *Projected GAN*.

Fig. 5 shows samples for a fixed noise vector **z** during training on LSUN-Church. For both FastGAN and StyleGAN, patches of texture gradually morph into a global structure. For Projected GAN, we directly observe the emergence of structure which becomes more detailed over time. Interestingly, the Projected GAN latent space appears to be very volatile, i.e., for fixed **z** the images undergo significant perceptual changes during training. In the non-projected cases, these changes are more gradual. We hypothesize that this induced volatility might be due to the discriminator providing more *semantic* feedback compared to conventional RGB losses. Such semantic feedback could introduce

more stochasticity during training which in turn improves convergence and performance. We also observed that the signed real logits of the discriminator remain at the same level throughout training (see Appendix). Stable signed logits indicate that the discriminator does not suffer from overfitting.

**Sample Efficiency.** The use of pretrained models is generally linked to improved sample efficiency. To evaluate this property, we also created two subsets of the 70k CLEVR dataset by randomly subsampling 10k and 1k images from it, respectively. As depicted in Fig. 4 (right), our Projected GAN significantly improves over both baselines across all dataset splits.

## 5.2 Large Datasets

Besides CLEVR and LSUN-Church, we benchmark Projected GANs against various state-of-the-art models on three other large datasets: LSUN-Bedroom [67] (3M indoor bedroom scenes), FFHQ [26] (70k images of faces) and Cityscapes [6] (25k driving scenes captured from a vehicle). For all datasets, we use an image resolution of $256^2$ pixels. As Cityscapes and CLEVR images are not of aspect ratio 1:1 we resize them to $256^2$ for training. Besides StyleGAN2-ADA and FastGAN, we compare against SAGAN [69] and GANsformers [20]. All models were trained for 10 M images. For the large datasets, we also report numbers for StyleGAN2 trained for more than 10 M images to report the lowest FID values achieved in previous literature (denoted as StyleGAN2*). In the appendix, we report results on nine more large datasets.

Table 3 shows that the Projected GAN outperforms all state-of-the-art models in terms of FID values on all datasets by a large margin. For example, on LSUN-Bedroom, it achieves an FID value of 1.52 compared to 6.15 by GANsformer, the previously best model in this setting. Projected GAN achieves state-of-the-art FID values remarkably fast, e.g., on LSUN-church, it achieves an FID value of 3.18 after 1.1 M *Imgs*. StyleGAN2 has obtained the previously lowest FID value of 3.39 after 88 M *Imgs*, 80 times as many as needed by Projected GAN. Similar speed-ups are also realized for all other large datasets as shown in Table 3. Interestingly, when training longer on FFHQ (39 M *Imgs*), we observe further improvements of Projected GAN to an FID of 2.2. Note that all five datasets represent very different objects in various scenes. This demonstrates that the performance gain is robust to the choice of the dataset, although the feature network is trained only on ImageNet. It is important to note that the main improvements are based on improved sample diversity as indicated by recall which we report in the appendix. The improvement in diversity is most notable on large datasets, e.g., LSUN church, where the image fidelity appears to be similar to StyleGAN.

## 5.3 Small Datasets

To further evaluate our method in the few-shot setting, we compare against StyleGAN2-ADA and FastGAN on art paintings from WikiArt (1000 images; wikiart.org), Oxford Flowers (1360 images) [42], photographs of landscapes (4319 images; flickr.com), AnimalFace-Dog (389 images) [57] and Pokemon (833 images; pokemon.com). Further, we report results on high-resolution versions of Pokemon and Art-Painting ($1024^2$). Lastly, we evaluate on AFHQ-Cat, -Dog and -Wild at $512^2$ [5]. The AFHQ datasets contain ~5k closeups per category cat, dog, or wildlife. We do not have a license to re-distribute these datasets, but we provide the URLs to enable reproducibility, similar to [35].

Projected GAN outperforms all baselines in terms of FID values by a significant margin on all datasets and all resolutions as shown in Table 3. Remarkably, our model beats the prior state-of-the-art on all datasets ($256^2$) after observing fewer than 600k images. For AnimalFace-Dog, the Projected GAN surpasses the previously best FID after only 20k images. One might argue that the EfficientNet used as feature network facilitates data generation for the animal datasets as EfficientNet is trained on ImageNet which contains many animal classes (e.g., 120 classes for dog breeds). However, it is interesting to observe that Projected GANs also achieve state-of-the-art FID on Pokemon and Art Painting though these datasets differ significantly from ImageNet. This evidences the generality of ImageNet features. For the high-resolution datasets, Projected GANs achieve the same FID value many times faster than the best baselines, e.g., ten times faster than StyleGAN2-ADA on AFHQ-Cat or four times faster than FastGAN on Pokemon. We remark that $F$ and $D_l$ generalize to any resolution as they are fully convolutional.

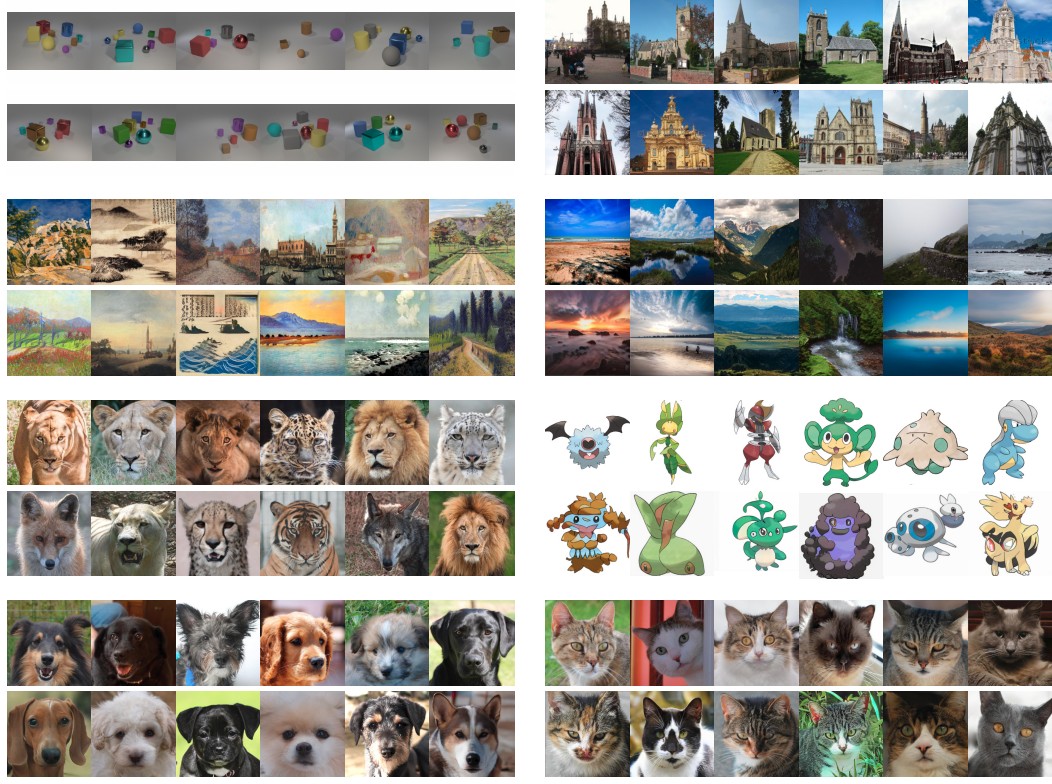

Figure 6: **Real samples (top rows) vs. samples by Projected GAN (bottom rows).** Datasets (top left to bottom right): CLEVR ($256^2$), LSUN church ($256^2$), Art Painting ($256^2$), Landscapes ($256^2$), AFHQ-wild ($512^2$), Pokemon ($256^2$), AFHQ-dog ($512^2$), AFHQ-cat ($512^2$).

| | FID | *Imgs* | FID | *Imgs* | FID | *Imgs* | FID | *Imgs* | FID | *Imgs* |
|---|---|---|---|---|---|---|---|---|---|---|
| | Large Datasets ($256^2$) | | | | | | | | | |
| | **CLEVR** | | **FFHQ** | | **Cityscapes** | | **Bedroom** | | **Church** | |
| SAGAN [69] | 26.04 | 10 M | 16.21 | 10 M | 12.81 | 10 M | 14.06 | 10 M | 6.15 | 10 M |
| STYLEGAN2-ADA [25] | 10.17 | 10 M | 7.32 | 10 M | 8.35 | 10 M | 11.53 | 10 M | 5.85 | 10 M |
| GANSFORMERS [20] | 9.24 | 10 M | 7.42 | 10 M | 5.23 | 10 M | 6.15 | 10 M | 5.47 | 10 M |
| FASTGAN [35] | 3.24 | 10 M | 12.69 | 10 M | 8.78 | 1.8 M | 8.24 | 4.8 M | 8.43 | 8.9 M |
| PROJECTED GAN | **0.89** | 4.5 M | **3.39** | 7.1 M | **3.41** | 1.7 M | **1.52** | 5.2 M | **1.59** | 9.2 M |
| PROJECTED GAN* | 3.39 | **0.5 M** | 3.56 | **7.0 M** | 4.60 | **1.1 M** | 2.58 | **1.5 M** | 3.18 | **1.1 M** |
| STYLEGAN2* [25, 26, 68] | 5.05 | 25 M | 3.62 | 25 M | - | - | 2.65 | 70 M | 3.39 | 88 M |
| | Small Datasets ($256^2$) | | | | | | | | | |
| | **Art Painting** | | **Landscape** | | **AnimalFace** | | **Flowers** | | **Pokemon** | |
| STYLEGAN2-ADA [25] | 43.07 | 3.2 M | 15.99 | 6.3 M | 60.90 | 2.2 M | 21.66 | 3.8 M | 40.38 | 3.4 M |
| FASTGAN [35] | 44.02 | 0.7 M | 16.44 | 1.8 M | 62.11 | 0.2 M | 26.23 | 0.8 M | 81.86 | 2.5 M |
| PROJECTED GAN | **27.96** | 0.8 M | **6.92** | 3.5 M | **17.88** | 10 M | **13.86** | 1.8 M | **26.36** | 0.8 M |
| PROJECTED GAN* | 40.22 | **0.2 M** | 14.99 | **0.6 M** | 58.07 | **0.02 M** | 21.60 | **0.2 M** | 36.57 | **0.3 M** |
| | $1024^2$ | | | | $512^2$ | | | | | |
| | **Art Painting** | | **Pokemon** | | **AFHQ-Cat** | | **AFHQ-Dog** | | **AFHQ-Wild** | |
| STYLEGAN2-ADA [25] | 41.69 | 1.0 M | 56.76 | 0.6 M | 3.55 | 10 M | 7.40 | 10 M | 3.05 | 10 M |
| FASTGAN [35] | 46.71 | 0.8 M | 56.46 | 0.8 M | 4.69 | 1.1 M | 13.09 | 1.6 M | 3.14 | **1.6 M** |
| PROJECTED GAN | **32.07** | 0.9 M | **33.96** | 1.3 M | **2.16** | 3.7 M | **4.52** | 3.8 M | **2.17** | 5.4 M |
| PROJECTED GAN* | 40.33 | **0.2 M** | 53.74 | **0.2 M** | 3.53 | **1.0 M** | 7.10 | **0.9 M** | 3.03 | **1.6 M** |

Table 3: **Quantitative Results.** Projected GAN* reports the point where our approach surpasses the state-of-the-art. StyleGAN2* obtains the lowest FID in previous literature if trained long enough.

# 6 Discussion and Future Work

While we achieve low FID on all datasets, we also identify two systematic failure cases: As depicted in Fig. 7, we sometimes observe "floating heads" on AFHQ. In a few samples, the animals appear in high quality but resemble cutouts on blurry or bland backgrounds. We hypothesize that generating a realistic background and image composition is less critical when a prominent object is already depicted. This hypothesis follows from the fact that we used image classification models for the projection, which have been shown to only marginally reduce in accuracy when applied on images of objects with removed background [66]. On FFHQ, projected GAN sometimes produces poor-quality samples with wrong proportions and artifacts, even at state-of-the-art FID, see Fig. 8.

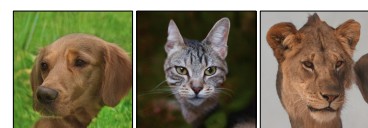

Figure 7: **"Floating Heads"**

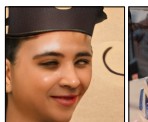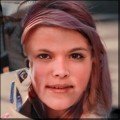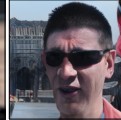

Figure 8: **Artifacts on FFHQ.**

In terms of generators, StyleGAN is more challenging to tune and does not profit as much from projected training. The FastGAN generator is fast to optimize but simultaneously produced unrealistic samples in some parts of the latent space – a problem that could be solved by a mapping network similar to StyleGAN. Hence, we speculate that unifying the strengths of both architectures in combination with projected training might improve performance further. Moreover, our study of different pretrained networks indicates that efficient models are especially suitable for projected GAN training. Exploring this connection in-depth, and in general, determining desirable feature space properties opens up exciting new research opportunities. Lastly, our work advances efficiency for generative models. More efficient models lower the barrier of computational effort needed for generating realistic images. A lower barrier facilitates malignant use of generative models (e.g., "deep fakes") while simultaneously also democratizing research in this area.

## Acknowledgments and Disclosure of Funding

We acknowledge the financial support by the BMWi in the project KI Delta Learning (project number 19A19013O). Andreas Geiger was supported by the ERC Starting Grant LEGO-3D (850533). Kashyap Chitta was supported by the German Federal Ministry of Education and Research (BMBF): Tübingen AI Center, FKZ: 01IS18039B and the International Max Planck Research School for Intelligent Systems (IMPRS-IS). Jens Müller received funding by the Heidelberg Collaboratory for Image Processing (HCI). We thank the Center for Information Services and High Performance Computing (ZIH) at Dresden University of Technology for generous allocations of computation time. Lastly, we would like to thank Vanessa Sauer for her general support.

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
