# Supplementary Material for
# Projected GANs Converge Faster

**Axel Sauer**[1,2]     **Kashyap Chitta**[1,2]     **Jens Müller**[3]     **Andreas Geiger**[1,2]

[1]University of Tübingen     [2]Max Planck Institute for Intelligent Systems, Tübingen

[3]Computer Vision and Learning Lab, University Heidelberg

[2]{firstname.lastname}@tue.mpg.de     [3]{firstname.lastname}@iwr.uni-heidelberg.de

In this **supplementary document**, we first prove the theorem presented in the paper in Section 1. Section 2 provides additional evaluation metrics for StyleGAN-ADA [12], FastGAN [20], and Projected GAN, and FID of Projected GAN on nine more datasets. Section 3 presents uncurated samples for both baselines and our approach. Section 4 reports additional experiments. Lastly, we provide details on training configurations, hyperparameters, and compute in Section 5. The **supplementary videos** show interpolations between random samples of Projected GAN on all datasets. Code, models, and supplementary videos can be found on the project page https://sites.google.com/view/projected-gan.

## 1 Proofs

As described in the paper, Projected GAN training can be formulated as follows

$$\min_G \max_{\{D_l\}} \sum_{l \in \mathcal{L}} \Big( \mathbb{E}_{\mathbf{x}}[\log D_l(P_l(\mathbf{x}))] + \mathbb{E}_{\mathbf{z}}[\log(1 - D_l(P_l(G(\mathbf{z}))))] \Big) \tag{1}$$

where $\{D_l\}$ is a set of independent discriminators operating on different feature projections. In the following, we first prove Theorem 1 for a deterministic projection. The second proof demonstrates the theorem's validity when training with stochastic differentiable augmentations.

*Proof (deterministic).* The following proof follows the consistency proofs in [23] and [7]. Let $\{P_l\}_{l \in \mathcal{L}}$ be a set of fixed feature projectors. Furthermore, let $\mathbb{P}_T$ be the density of the true data distribution and $\mathbb{P}_G$ the density of the distribution the generator $G$ produces. As in Theorem 1, $P_l \circ T$ and $P_l \circ G$ are functional compositions of $P_l$ and the true/generated data distribution. The minimax objective in (1) is then defined via

$$\min_G \max_{\{D_l\}} \sum_{l \in \mathcal{L}} V_l(D_l, G) \tag{2}$$

where

$$\begin{aligned} V_l\left(D_l, G\right) &= \mathbb{E}_{\mathbf{x} \sim \mathbb{P}_T}\left[\log D_l\left(P_l(\mathbf{x})\right)\right] + \mathbb{E}_{\mathbf{x} \sim \mathbb{P}_G}\left[\log\left(1 - D_l\left(P_l(\mathbf{x})\right)\right)\right] \\ &= \mathbb{E}_{\mathbf{y} \sim \mathbb{P}_{P_l \circ T}}\left[\log D_l(\mathbf{y})\right] + \mathbb{E}_{\mathbf{y} \sim \mathbb{P}_{P_l \circ G}}\left[\log\left(1 - D_l(\mathbf{y})\right)\right] \\ &= \int_{\mathbf{y}} \mathbb{P}_{P_l \circ T}(\mathbf{y}) \log(D_l(\mathbf{y})) + \mathbb{P}_{P_l \circ G}(\mathbf{y}) \log(1 - D_l(\mathbf{y})) d\mathbf{y} \end{aligned} \tag{3}$$

In the following we derive the optimal discriminator for a fixed $G$. For any $(a, b) \in \mathbb{R}^2 \setminus \{(0, 0)\}$, the function $t \to a \log(t) + b \log(1 - t)$ obtains its maximum in $[0, 1]$ at $\frac{a}{a+b}$ [7]. Since the discriminators do not need to be defined outside $\operatorname{supp}\left(\mathbb{P}_{P_l \circ T}\right) \cup \operatorname{supp}\left(\mathbb{P}_{P_l \circ G}\right)$, the maximum $\max_{\{D_l\}} V_l(D_l, G)$ is achieved for

$$D_{l,G}^*(\boldsymbol{y}) = \frac{\mathbb{P}_{P_l \circ T}(\mathbf{y})}{\mathbb{P}_{P_l \circ T}(\mathbf{y}) + \mathbb{P}_{P_l \circ G}(\mathbf{y})} \tag{4}$$

where $G$ is fixed. Assuming a perfect discriminator, the minimax objective can be reformulated as

$$C(G) = \max_{\{D_l\}} \sum_l V_l(G, D_l) = \sum_l V_l(G, D_{l,G}^*) \tag{5}$$

35th Conference on Neural Information Processing Systems (NeurIPS 2021).

Following the arguments of [7] and in [23], we obtain

$$C(G) = -2|\mathcal{L}|\log(2) + 2\sum_l JSD(\mathbb{P}_{P_l \circ T} || \mathbb{P}_{P_l \circ G}) \tag{6}$$

where $JSD$ is the Jensen-Shannon divergence. Since the Jensen-Shannon divergence is non-negative and zero only in case of equality, the minimum is achieved iff $\mathbb{P}_{P_l \circ T} = \mathbb{P}_{P_l \circ G}$ for all $l$. This shows, that we achieve $\min_G C(G)$ iff $\mathbb{P}_{P_l \circ T} = \mathbb{P}_{P_l \circ G}$ for all $l$.

*Proof (stochastic).* We now show that the result above still holds when applying stochastic differentiable augmentations before the feature projections. Utilizing a stochastic augmentation $f_{\boldsymbol{\theta},l}$ before the projection through $P_l$, can be viewed as a functional composition, i.e. $P_{\boldsymbol{\theta},l} = P_l \circ f_{\boldsymbol{\theta},l}$. The parameter $\boldsymbol{\theta} \sim \mathbb{P}_\Theta$ encompasses both the probability of applying the augmentation and its parameters, e.g., translation direction and magnitude. As in the deterministic case, the minimax objective is defined as

$$\min_G \max_{\{D_l\}} \sum_{l \in \mathcal{L}} V_l(D_l, G) \tag{7}$$

where

$$
\begin{aligned}
V_l(D_l, G) &= \mathbb{E}_{\mathbf{x} \sim \mathbb{P}_T}\left[\mathbb{E}_{\boldsymbol{\theta} \sim \mathbb{P}_\Theta}[\log D_l(P_{\boldsymbol{\theta},l}(\mathbf{x}))]\right] + \mathbb{E}_{\mathbf{x} \sim \mathbb{P}_G}\left[\mathbb{E}_{\boldsymbol{\theta} \sim \mathbb{P}_\Theta}[\log(1 - D_l(P_{\boldsymbol{\theta},l}(\mathbf{x})))]\right] \\
&= \mathbb{E}_{\boldsymbol{\theta} \sim \mathbb{P}_\Theta}\left[\mathbb{E}_{\mathbf{x} \sim \mathbb{P}_T}[\log D_l(P_{\boldsymbol{\theta},l}(\mathbf{x}))] + \mathbb{E}_{\mathbf{x} \sim \mathbb{P}_G}[\log(1 - D_l(P_{\boldsymbol{\theta},l}(\mathbf{x})))]\right] \\
&= \mathbb{E}_{\boldsymbol{\theta} \sim \mathbb{P}_\Theta}\left[\mathbb{E}_{\mathbf{y} \sim \mathbb{P}_{P_{\boldsymbol{\theta},l} \circ T}}[\log D_l(\mathbf{y})] + \mathbb{E}_{\mathbf{y} \sim \mathbb{P}_{P_{\boldsymbol{\theta},l} \circ G}}[\log(1 - D_l(\mathbf{y}))]\right] \\
&= \mathbb{E}_{\boldsymbol{\theta} \sim \mathbb{P}_\Theta}\left[\int_{\mathbf{y}} \mathbb{P}_{P_{\boldsymbol{\theta},l} \circ T}(\mathbf{y})\log(D_l(\mathbf{y})) + \mathbb{P}_{P_{\boldsymbol{\theta},l} \circ G}(\mathbf{y})\log(1 - D_l(\mathbf{y}))d\mathbf{y}\right] \\
&= \int_{\mathbf{y}} \mathbb{E}_{\boldsymbol{\theta} \sim \mathbb{P}_\Theta}[\mathbb{P}_{P_{\boldsymbol{\theta},l} \circ T}(\mathbf{y})]\log(D_l(\mathbf{y})) + \mathbb{E}_{\boldsymbol{\theta} \sim \mathbb{P}_\Theta}[\mathbb{P}_{P_{\boldsymbol{\theta},l} \circ G}(\mathbf{y})]\log(1 - D_l(\mathbf{y}))d\mathbf{y}
\end{aligned} \tag{8}
$$

Using the same arguments as above, we obtain that the maximum $\max_{\{D_l\}} V_l(D_l, G)$ is achieved for

$$D_{l,G}^*(\boldsymbol{y}) = \frac{\mathbb{E}_{\boldsymbol{\theta} \sim \mathbb{P}_\Theta}[\mathbb{P}_{P_{\boldsymbol{\theta},l} \circ T}(\mathbf{y})]}{\mathbb{E}_{\boldsymbol{\theta} \sim \mathbb{P}_\Theta}[\mathbb{P}_{P_{\boldsymbol{\theta},l} \circ T}(\mathbf{y})] + \mathbb{E}_{\boldsymbol{\theta} \sim \mathbb{P}_\Theta}[\mathbb{P}_{P_{\boldsymbol{\theta},l} \circ G}(\mathbf{y})]} \tag{9}$$

where $G$ is fixed. Note that $\mathbb{E}_{\boldsymbol{\theta} \sim \mathbb{P}_\Theta}[\mathbb{P}_{P_{\boldsymbol{\theta},l} \circ T}]$ and $\mathbb{E}_{\boldsymbol{\theta} \sim \mathbb{P}_\Theta}[\mathbb{P}_{P_{\boldsymbol{\theta},l} \circ G}]$ are densities. Similar to above, we obtain $\min_G C(G)$ iff $\mathbb{E}_{\boldsymbol{\theta} \sim \mathbb{P}_{\Theta,l}}[\mathbb{P}_{P_l \circ T}] = \mathbb{E}_{\boldsymbol{\theta} \sim \mathbb{P}_\Theta}[\mathbb{P}_{P_{\boldsymbol{\theta},l} \circ G}]$ for all $l$.

## 2 Additional Metrics and Datasets

**Metrics.** In addition to FID and *Imgs* reported in the paper, we compute the following metrics:

- **Kernel Inception Distance (KID) [2].** KID is an unbiased alternative to FID, hence, especially suitable for small datasets.

- **Precision and Recall [28].** Precision measures the quality of samples, and recall measures image diversity. Generally, GANs produce high-quality samples while being prone to mode collapse (high precision, low recall), compared to VAEs [16], which generate lower quality but more diverse samples (low precision, high recall). This observation is evidenced empirically in [28]. For our evalution, we use the improved formulation by [18].

- **SwAV-FID [22].** Instead of utilizing an image classifier feature space, SwAV-FID uses self-supervised representations. More specifically, SwAV-FID computes the Fréchet distance in the penultimate layer of a ResNet-50 trained with the contrastive SwAV objective [3]. Morozov et al. [22] show that in some cases, SwAV-FID is more consistent with human judgment of visual quality than FID.

- **CLIP-FID & Virtex-FID.** FID uses an Inception network trained on ImageNet. Our feature network $F$ has also been trained on ImageNet. To rule out training data as a confounding factor between $F$ and the evaluation metric, we propose to use FID with non-ImageNet features. We evaluate CLIP-FID (using a ResNet50 trained with the CLIP objective on the dataset collected by [25]) and VirTex-FID (using a ResNet50 trained on COCO Captions with the VirTex objective [5]).

- **Sliced Wasserstein Distance (SWD).** SWD is a non-neural metric that computes the Wasserstein distance between local image patches drawn from a Laplacian pyramid. We follow the evaluation protocol proposed by [11].

In addition to the metrics above, we conduct a **human preference study** with 28 participants unfamiliar with our method. The structure of the study is as follows:

- For each dataset, we first show samples of the real dataset for context.
- We then present two sample sheets and ask the participants to rank the sheets relative to each other based on sample fidelity and diversity. We define these as follows: (i) Fidelity is the degree to which the generated samples resemble the real ones. (ii) Diversity measures whether the generated samples cover the full variability of the real samples.
- Each sheet contains nine random samples; we present three sample sheet pairs per dataset.
- Samples and comparison pairs are randomized per participant. We make sure that all possible pairings are equally represented.
- Evaluated Models: StyleGAN2-ADA, FastGAN, Projected GAN, and real data (for control)
- Evaluated datasets: all 256x256 datasets
- We count how often a given model wins a comparison and report the relative amount of wins.

**Results.** On all datasets, KID, SwaV-FID, CLIP-FID, VirTex-FID, and SWD mirror the ranking obtained via FID. The low SwAV-FIDs indicate that Projected GAN's low FIDs are not due to correlations between the feature network used for projection and the inception network used in FID.

On the large datasets, the baselines only outperform Projected GAN in precision on FFHQ, Cityscapes, and LSUN Church (Table 1). However, when comparing recall in these cases, it is apparent that the baselines suffer from mode collapse. The high recall generally obtained by Projected GAN hints at the reason for its superior performance on FID, KID, and SwAV-FID: the generated images are very diverse. Hence, we conclude that Projected GANs alleviate mode collapse. The sample diversity is also evident in the qualitative comparisons in Section 3. On small datasets (Table 2), Projected GAN is outperformed in precision on Art Painting by FastGAN; however, FastGANs very low recall of 0.044 hints at mode collapse. Only on flowers, Projected GAN appears to cover fewer modes than the baselines as indicated by lower recall.

At high resolutions (Table 3), Projected GAN performs slightly worse in precision. It appears that Projected GAN incurs small losses in image quality while obtaining a better mode coverage, which can be observed in the quantitative comparisons, e.g., on AFHQ-Cat, some samples exhibit artifacts. These artifacts indicate that projected GAN training at higher resolutions warrants closer inspection.

On small datasets, overfitting is a problem that is not detected well by FID and other metrics [27]. Therefore, it is instructive to inspect latent interpolations for which we refer the reader to the supplementary videos. Projected GAN generates smooth interpolations between random samples on all datasets suggesting that it generalizes rather than memorizing training samples.

The results of the human preference study are shown in Table 4. The study results largely agree with the results obtained via FID. On FFHQ, the study demonstrates our reported failure case for projected GANs. Interestingly, on AnimalFace projected GAN outperforms real data. We hypothesize that this is because for AnimalFace there is a significant portion of low-quality images (blurry, compression artifacts) in the dataset, and possibly projected GAN generates fewer of those. Of course, human studies are not optimal, as it is not straightforward to evaluate sample diversity - which is a strong suit of projected GANs - given only a few samples.

Table 5 reports the FID achieved by Projected GAN for nine more datasets, all at a resolution of $256^2$. We compare on LSUN cat and horse [31], ADE indoor (a subset of ADE [34] proposed in [1]), the full Oxford flowers dataset with 8k images [24], KITTI-fisheye (a subset of KITTI-360 [19], consisting of fisheye images), STL-10 [4], CUB200 [30], Stanford Dogs [14], and Stanford Cars [17]. We do not change the hyperparameters of Projected GAN. On each dataset, we report the lowest FID achieved in previous literature. We train FastGAN as a baseline for ADE indoor and KITTI-fisheye.

| | Large Datasets ($256^2$) | | | | |
|---|---|---|---|---|---|
| | **CLEVR** | **FFHQ** | **Cityscapes** | **Bedroom** | **Church** |
| | $FID \downarrow$ | | | | |
| STYLEGAN2-ADA [12] | 10.17 | 7.32 | 8.35 | 11.53 | 5.85 |
| FASTGAN [20] | 3.24 | 12.69 | 8.78 | 8.24 | 8.43 |
| PROJECTED GAN | **0.89** | **3.08** | **3.41** | **1.52** | **1.59** |
| | $KID \times 10^3 \downarrow$ | | | | |
| STYLEGAN2-ADA [12] | 8.15 | 1.49 | 3.34 | 7.42 | 4.70 |
| FASTGAN [20] | 2.64 | 5.34 | 5.45 | 5.90 | 4.61 |
| PROJECTED GAN | **0.51** | **0.44** | **0.91** | **0.36** | **0.50** |
| | $Precision \uparrow$ | | | | |
| STYLEGAN2-ADA [12] | 0.373 | 0.669 | **0.649** | 0.429 | 0.565 |
| FASTGAN [20] | 0.600 | **0.716** | 0.557 | 0.602 | **0.645** |
| PROJECTED GAN | **0.640** | 0.654 | 0.619 | **0.614** | 0.612 |
| | $Recall \uparrow$ | | | | |
| STYLEGAN2-ADA [12] | 0.569 | 0.445 | 0.146 | 0.202 | 0.416 |
| FASTGAN [20] | 0.650 | 0.184 | 0.227 | 0.189 | 0.207 |
| PROJECTED GAN | **0.735** | **0.464** | **0.361** | **0.346** | **0.438** |
| | $SwAV - FID \downarrow$ | | | | |
| STYLEGAN2-ADA [12] | 3.50 | 1.24 | 1.35 | 8.47 | 2.51 |
| FASTGAN [20] | 1.46 | 2.55 | 1.29 | 5.38 | 3.64 |
| PROJECTED GAN | **0.56** | **0.85** | **0.60** | **1.44** | **1.01** |
| | $CLIP - FID \downarrow$ | | | | |
| STYLEGAN2-ADA [12] | 4.70 | 10.3 | 5.88 | 42.12 | 15.85 |
| FASTGAN [20] | 4.24 | 19.23 | 6.46 | 31.10 | 35.47 |
| PROJECTED GAN | **0.80** | **7.55** | **2.96** | **11.97** | **13.71** |
| | $VirTex - FID \downarrow$ | | | | |
| STYLEGAN2-ADA [12] | 0.78 | 1.20 | 1.15 | 2.20 | 1.10 |
| FASTGAN [20] | 0.64 | 2.47 | 1.48 | 2.66 | 3.61 |
| PROJECTED GAN | **0.35** | **0.64** | **0.49** | **0.81** | **0.82** |
| | $SWD \times 10^{-3} \downarrow$ | | | | |
| STYLEGAN2-ADA [12] | 17.50 | 7.42 | 10.71 | 12.53 | 14.62 |
| FASTGAN [20] | 28.51 | 10.19 | 9.45 | 14.68 | 14.42 |
| PROJECTED GAN | **12.90** | **6.41** | **7.27** | **6.83** | **8.37** |

Table 1: **Metrics on Large Datasets** ($256^2$). Projected GAN compares favorably on most metrics. Exceptions are precision on FFHQ, Cityscapes, and LSUN Church. As argued by [13], shifting from precision to recall is generally desirable, since recall can be traded into precision via truncation.

| | Small Datasets ($256^2$) | | | | |
|---|---|---|---|---|---|
| | **Art Painting** | **Landscape** | **AnimalFace** | **Flowers** | **Pokemon** |
| | | | $FID \downarrow$ | | |
| STYLEGAN2-ADA [12] | 43.07 | 15.99 | 60.90 | 21.66 | 40.38 |
| FASTGAN [20] | 44.02 | 16.44 | 62.11 | 26.23 | 81.86 |
| PROJECTED GAN | **27.96** | **6.92** | **17.88** | **13.86** | **26.36** |
| | | | $KID \times 10^3 \downarrow$ | | |
| STYLEGAN2-ADA [12] | 10.23 | 4.39 | 22.52 | 3.56 | 13.49 |
| FASTGAN [20] | 13.00 | 3.40 | 22.11 | 6.61 | 80.30 |
| PROJECTED GAN | **1.25** | **1.30** | **0.03** | **0.38** | **1.32** |
| | | | $Precision \uparrow$ | | |
| STYLEGAN2-ADA [12] | 0.691 | 0.709 | 0.841 | 0.731 | 0.735 |
| FASTGAN [20] | **0.858** | 0.768 | 0.849 | 0.611 | 0.731 |
| PROJECTED GAN | 0.762 | **0.774** | **0.998** | **0.816** | **0.809** |
| | | | $Recall \uparrow$ | | |
| STYLEGAN2-ADA [12] | 0.218 | 0.213 | 0.036 | 0.095 | 0.197 |
| FASTGAN [20] | 0.044 | 0.160 | 0.015 | **0.100** | 0.004 |
| PROJECTED GAN | **0.239** | **0.258** | **0.095** | 0.058 | **0.259** |
| | | | $SwAV - FID \downarrow$ | | |
| STYLEGAN2-ADA [12] | 3.32 | 2.98 | 16.26 | 5.02 | 6.71 |
| FASTGAN [20] | 3.29 | 2.42 | 15.07 | 7.45 | 9.25 |
| PROJECTED GAN | **2.25** | **1.42** | **4.22** | **2.70** | **2.04** |
| | | | $CLIP - FID \downarrow$ | | |
| STYLEGAN2-ADA [12] | 44.13 | 24.89 | 46.18 | 26.30 | 13.96 |
| FASTGAN [20] | 40.47 | 19.84 | 54.69 | 40.12 | 87.65 |
| PROJECTED GAN | **22.91** | **13.71** | **16.89** | **15.83** | **9.93** |
| | | | $VirTex - FID \downarrow$ | | |
| STYLEGAN2-ADA [12] | 4.15 | 2.78 | 8.83 | 3.25 | 3.69 |
| FASTGAN [20] | 5.72 | 3.86 | 9.41 | 4.08 | 17.49 |
| PROJECTED GAN | **3.53** | **1.98** | **3.79** | **2.19** | **2.55** |
| | | | $SWD \times 10^{-3} \downarrow$ | | |
| STYLEGAN2-ADA [12] | 25.55 | 19.06 | 22.31 | 14.04 | 14.73 |
| FASTGAN [20] | 21.94 | 29.87 | 29.23 | 17.39 | 46.81 |
| PROJECTED GAN | **11.44** | **15.38** | **14.34** | **9.61** | **11.65** |

Table 2: **Metrics on Small Datasets** ($256^2$). Projected GAN performs best on most metrics.

| | 1024² | | 512² | | |
|---|---|---|---|---|---|
| | **Art Painting** | **Pokemon** | **AHFQ-Cat** | **AFHQ-Dog** | **AFHQ-Wild** |
| | *FID* ↓ | | | | |
| STYLEGAN2-ADA [12] | 41.69 | 56.76 | 3.55 | 7.40 | 3.05 |
| FASTGAN [20] | 46.71 | 56.46 | 4.69 | 13.09 | 3.14 |
| PROJECTED GAN | **32.07** | **33.96** | **2.16** | **4.52** | **2.17** |
| | *KID* × 10³ ↓ | | | | |
| STYLEGAN2-ADA [12] | 26.59 | 15.31 | 0.63 | 1.21 | 0.47 |
| FASTGAN [20] | 12.70 | 29.40 | 1.72 | 5.51 | 0.74 |
| PROJECTED GAN | **1.70** | **7.76** | **0.16** | **0.80** | **0.12** |
| | *Precision* ↑ | | | | |
| STYLEGAN2-ADA [12] | 0.619 | **0.791** | 0.767 | **0.753** | **0.765** |
| FASTGAN [20] | **0.776** | 0.777 | **0.784** | 0.746 | 0.761 |
| PROJECTED GAN | 0.706 | 0.780 | 0.693 | 0.718 | 0.705 |
| | *Recall* ↑ | | | | |
| STYLEGAN2-ADA [12] | 0.168 | 0.053 | 0.411 | 0.470 | 0.137 |
| FASTGAN [20] | 0.033 | 0.080 | 0.305 | 0.380 | 0.201 |
| PROJECTED GAN | **0.235** | **0.215** | **0.565** | **0.643** | **0.292** |
| | *SwAV − FID* ↓ | | | | |
| STYLEGAN2-ADA [12] | 3.68 | 5.03 | 1.23 | 1.98 | 1.89 |
| FASTGAN [20] | 3.41 | 5.14 | 1.73 | 3.07 | 1.77 |
| PROJECTED GAN | **2.28** | **3.83** | **0.68** | **1.12** | **1.08** |

Table 3: **Metrics on Small Datasets** ($512^2$ **and** $1024^2$**).** The results are in line with the findings at a resolution of $256^2$. Only with respect to precision, the baselines slightly outperform Projected GAN.

| | CLEVR | FFHQ | Cityscapes | Bedroom | Church | ArtPainting | Landscape | AnimalFace | Flowers | Pokemon | All Datasets |
|---|---|---|---|---|---|---|---|---|---|---|---|
| | Fidelity | | | | | | | | | | |
| STYLEGAN2-ADA [12] | 14 % | 32 % | 17 % | 5 % | 16 % | 16 % | 21 % | 4 % | 17 % | 10 % | 15 % |
| FASTGAN [20] | 7 % | 2 % | 15 % | 3 % | 9 % | 0 % | 8 % | 7 % | 4 % | 0 % | 6 % |
| PROJECTED GAN | 42 % | 5 % | 15 % | 16 % | 9 % | 28 % | 17 % | 55 % | 25 % | 38 % | 25 % |
| DATA | 37 % | 61 % | 53 % | 76 % | 66 % | 56 % | 54 % | 34 % | 54 % | 52 % | 54 % |
| | Diversity | | | | | | | | | | |
| STYLEGAN2-ADA [12] | 13 % | 24 % | 9 % | 10 % | 19 % | 11 % | 25 % | 9 % | 17 % | 6 % | 14 % |
| FASTGAN [20] | 13 % | 4 % | 11 % | 2 % | 0 % | 4 % | 13 % | 9 % | 17 % | 0 % | 7 % |
| PROJECTED GAN | 31 % | 12 % | 21 % | 21 % | 14 % | 27 % | 16 % | 49 % | 29 % | 40 % | 27 % |
| DATA | 43 % | 60 % | 59 % | 67 % | 67 % | 58 % | 46 % | 33 % | 37 % | 54 % | 52 % |

Table 4: **Human Preference Study.** The obtained results largely agree with the rankings of other metrics.

|  | LSUN cat | LSUN horse | ADE indoor | Flowers Full | KITTI fisheye |
|---|---|---|---|---|---|
| Prev. SotA | 5.57 | 2.57 | 30.33 | 19.60 | 6.64 |
| (Approach) | (ADM [6]) | (ADM [6]) | (FastGAN [20]) | (MSG-StyleGAN [10]) | (FastGAN [20]) |
| Projected GAN | **3.89** | **2.17** | **6.70** | **3.86** | **2.72** |

|  | STL-10 | CUB200 | Stanford Dogs | Stanford Cars |
|---|---|---|---|---|
| Prev. SotA | 25.32 | 11.25 | 25.66 | 16.03 |
| (Approach) | (TransGAN [9]) | (FineGAN [29]) | (FineGAN [29]) | (FineGAN [29]) |
| Projected GAN | **13.68** | **2.79** | **11.75** | **2.09** |

Table 5: **FID on Additional Datasets** ($256^2$) Without any hyperparameter changes, Projected GAN outperforms the previous state-of-the art on all evaluated datasets.

# 3 Qualitative Comparisons

We show generated images for CLEVR (Fig. 1), FFHQ (Fig. 2), Cityscapes (Fig. 3), Bedroom (Fig. 4), Church (Fig. 5), Art Painting (Fig. 6, 11), Landscape (Fig. 7), AnimalFace Dog (Fig. 8), Flowers (Fig. 9), Pokemon (Fig. 10, 12), AFHQ-Cat (Fig. 13), AFHQ-Dog (Fig. 14), and AFHQ-Wild (Fig. 15). Following [12], we select a global seed per dataset. We do not perform truncation on any of the models. Projected GAN produces convincing results on all datasets. The sample diversity in particular is apparent in comparison to the baselines, e.g., on AFHQ-Cat or AFHQ-Wild, all baselines generate high-quality samples, but Projected GAN captures more variability of the training data.

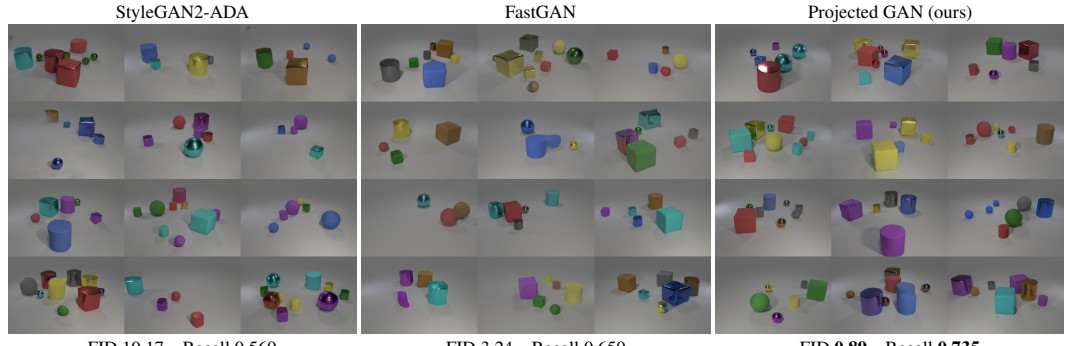

Figure 1: **Uncurated Results for CLEVR** ($256^2$). The images are selected randomly given one global random seed. We recommend zooming in for comparison.

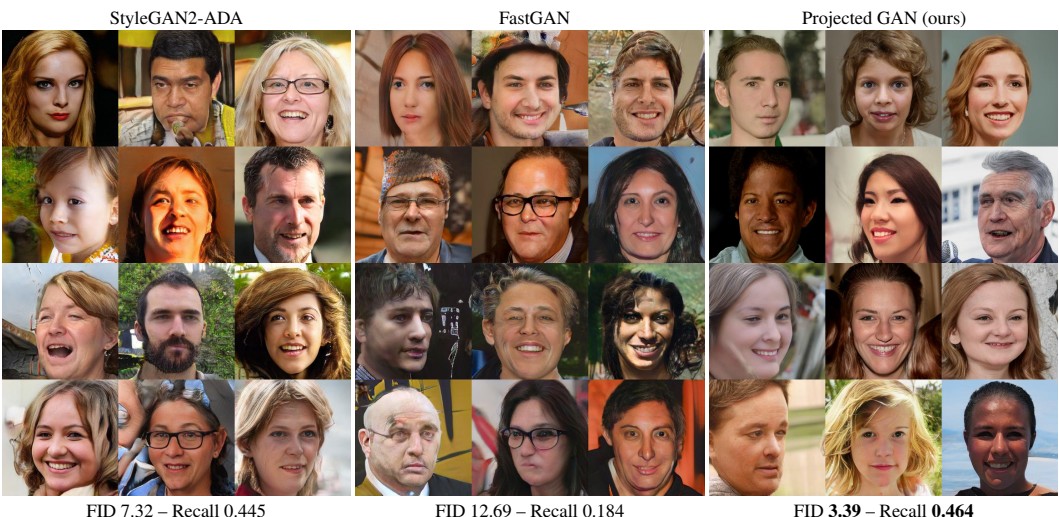

Figure 2: **Uncurated Results for FFHQ** ($256^2$). The images are selected randomly given one global random seed. We recommend zooming in for comparison.

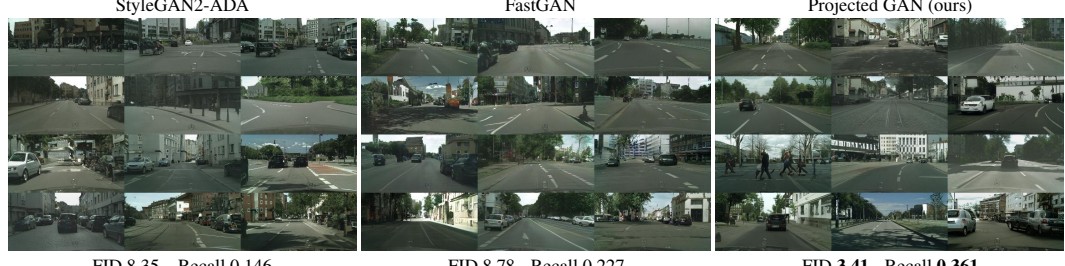

StyleGAN2-ADA        FastGAN        Projected GAN (ours)

FID 8.35 – Recall 0.146     FID 8.78 - Recall 0.227     FID **3.41** - Recall **0.361**

Figure 3: **Uncurated Results for Cityscapes** ($256^2$). The images are selected randomly given one global random seed. We recommend zooming in for comparison.

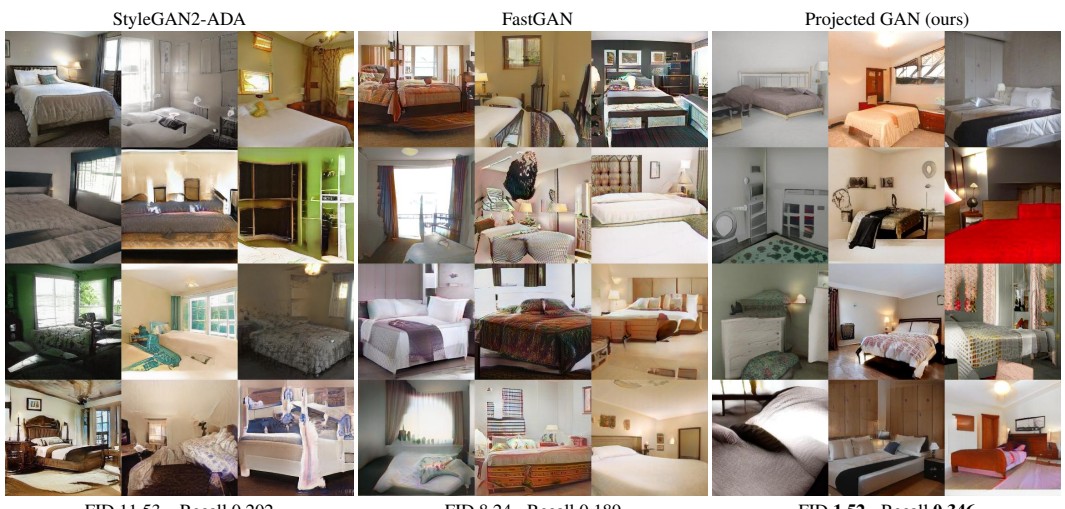

StyleGAN2-ADA        FastGAN        Projected GAN (ours)

FID 11.53 – Recall 0.202     FID 8.24 - Recall 0.189     FID **1.52** - Recall **0.346**

Figure 4: **Uncurated Results for LSUN bedroom** ($256^2$). The images are selected randomly given one global random seed. We recommend zooming in for comparison.

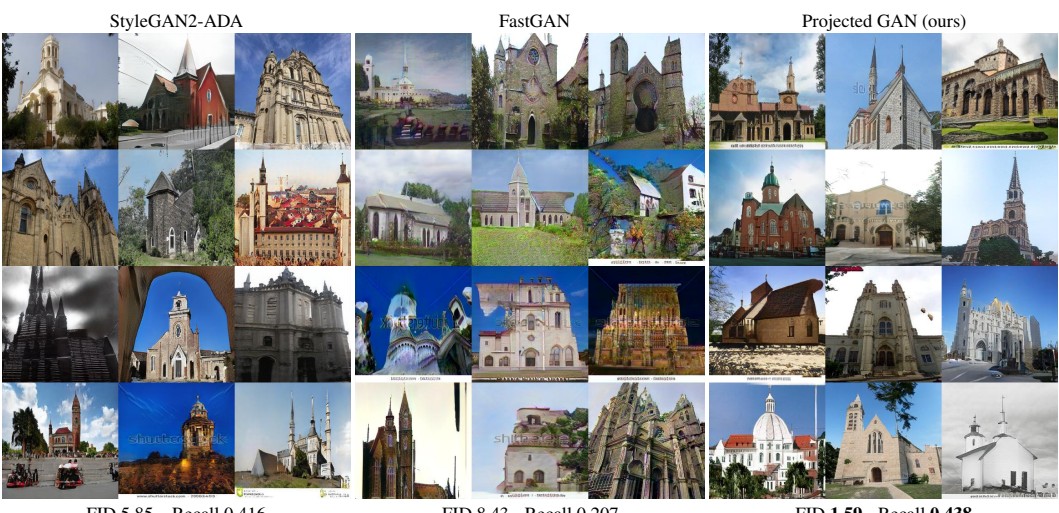

StyleGAN2-ADA        FastGAN        Projected GAN (ours)

FID 5.85 – Recall 0.416     FID 8.43 - Recall 0.207     FID **1.59** - Recall **0.438**

Figure 5: **Uncurated Results for LSUN church** ($256^2$). The images are selected randomly given one global random seed. We recommend zooming in for comparison.

StyleGAN2-ADA      FastGAN      Projected GAN (ours)

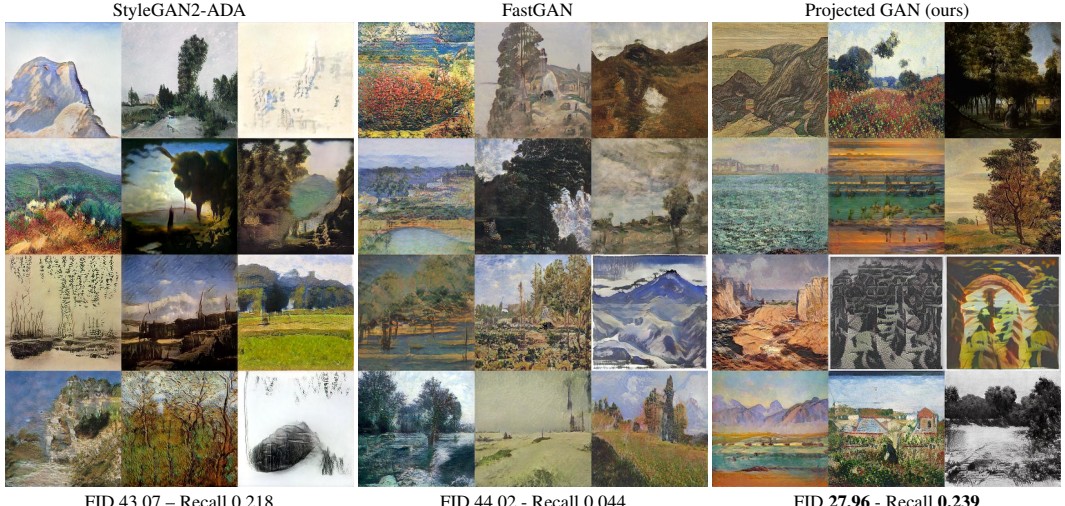

FID 43.07 – Recall 0.218    FID 44.02 - Recall 0.044    FID **27.96** - Recall **0.239**

Figure 6: **Uncurated Results for Art Painting** ($256^2$)**.** The images are selected randomly given one global random seed. We recommend zooming in for comparison.

StyleGAN2-ADA      FastGAN      Projected GAN (ours)

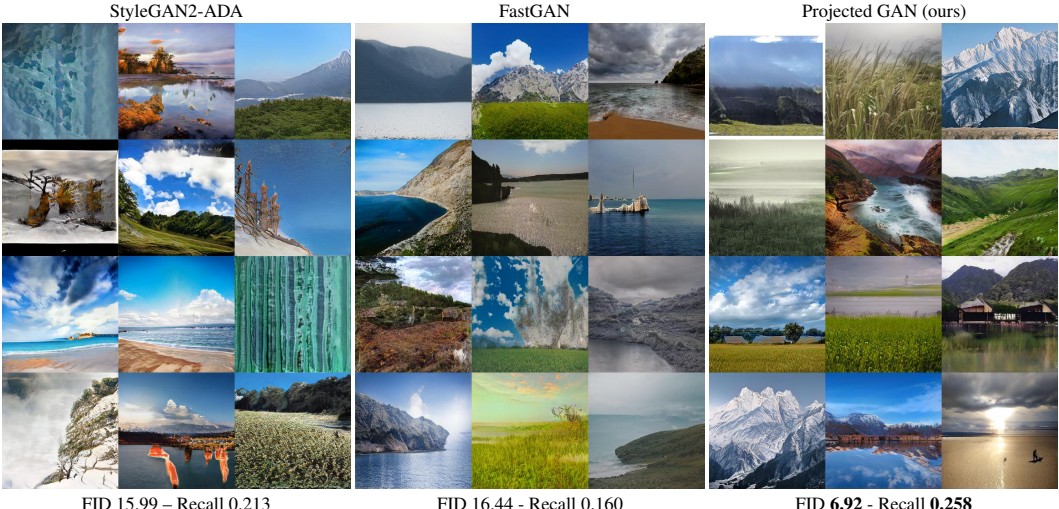

FID 15.99 – Recall 0.213    FID 16.44 - Recall 0.160    FID **6.92** - Recall **0.258**

Figure 7: **Uncurated Results for Landscape** ($256^2$)**.** The images are selected randomly given one global random seed. We recommend zooming in for comparison.

StyleGAN2-ADA                  FastGAN                  Projected GAN (ours)

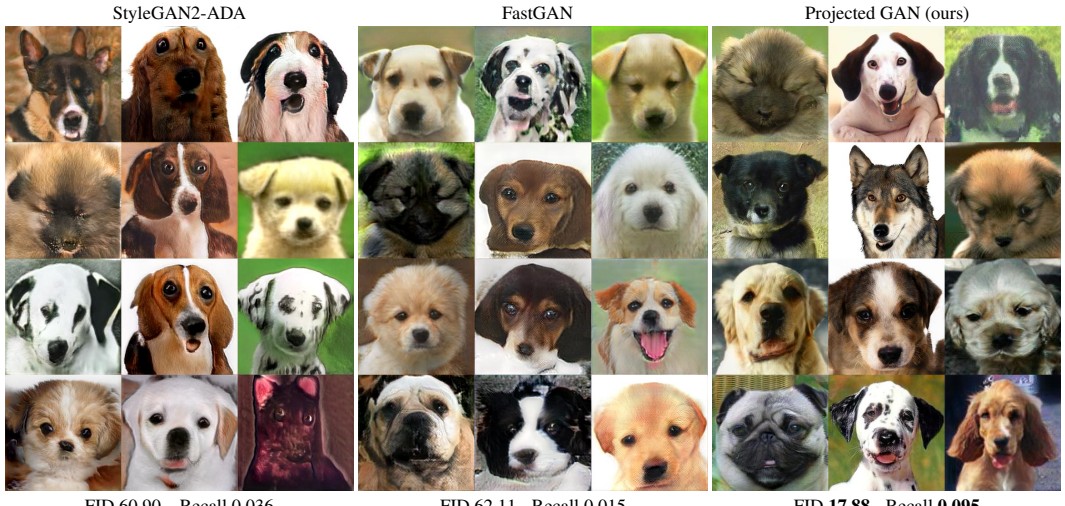

FID 60.90 – Recall 0.036        FID 62.11 - Recall 0.015        FID **17.88** - Recall **0.095**

Figure 8: **Uncurated Results for AnimalFace-Dog** ($256^2$). The images are selected randomly given one global random seed. We recommend zooming in for comparison.

StyleGAN2-ADA                  FastGAN                  Projected GAN (ours)

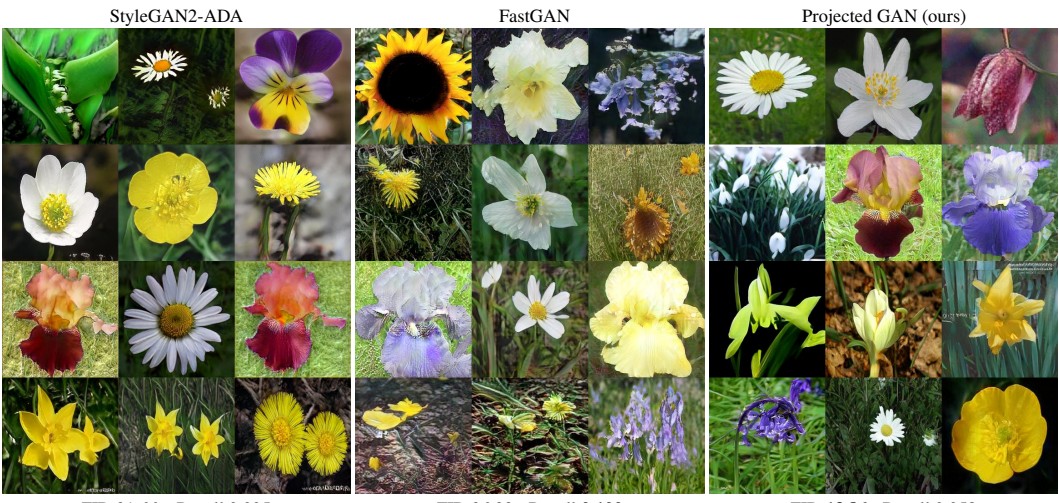

FID 21.66 – Recall 0.095        FID 26.23 - Recall **0.100**        FID **13.86** - Recall 0.058

Figure 9: **Uncurated Results for Flowers** ($256^2$). The images are selected randomly given one global random seed. We recommend zooming in for comparison.

StyleGAN2-ADA      FastGAN      Projected GAN (ours)

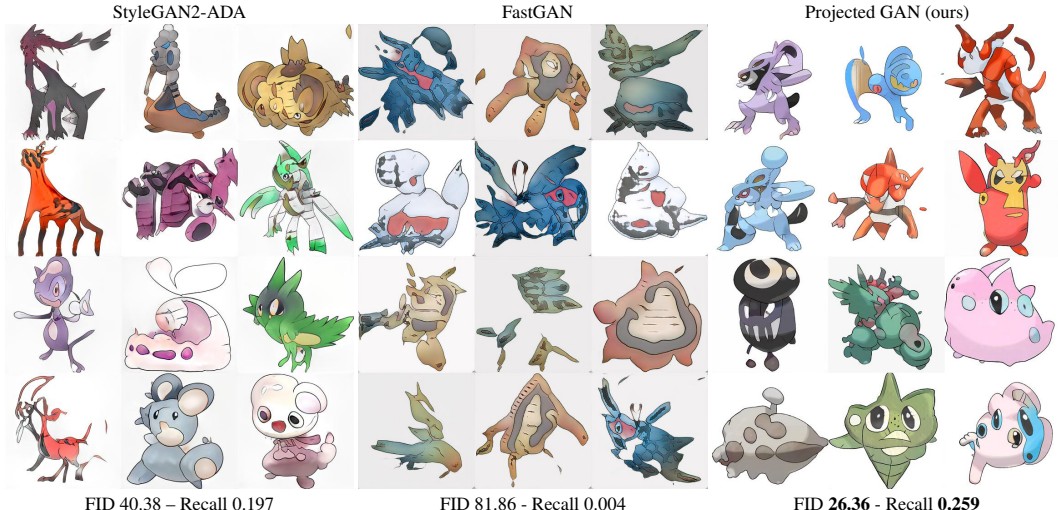

FID 40.38 – Recall 0.197   FID 81.86 - Recall 0.004   FID **26.36** - Recall **0.259**

Figure 10: **Uncurated Results for Pokemon** ($256^2$)**.** The images are selected randomly given one global random seed. We recommend zooming in for comparison.

StyleGAN2-ADA      FastGAN      Projected GAN (ours)

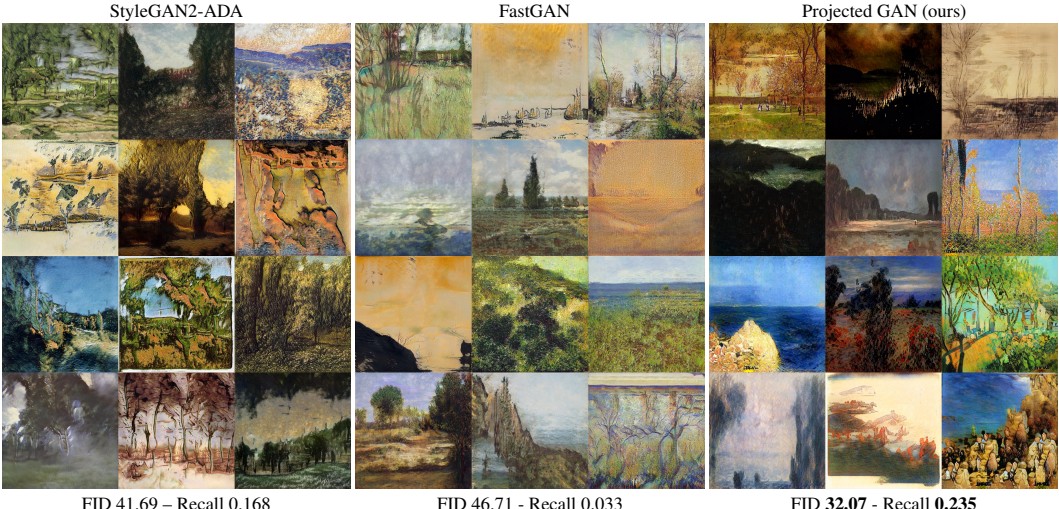

FID 41.69 – Recall 0.168   FID 46.71 - Recall 0.033   FID **32.07** - Recall **0.235**

Figure 11: **Uncurated Results for Art Painting** ($1024^2$)**.** The images are selected randomly given one global random seed. We recommend zooming in for comparison.

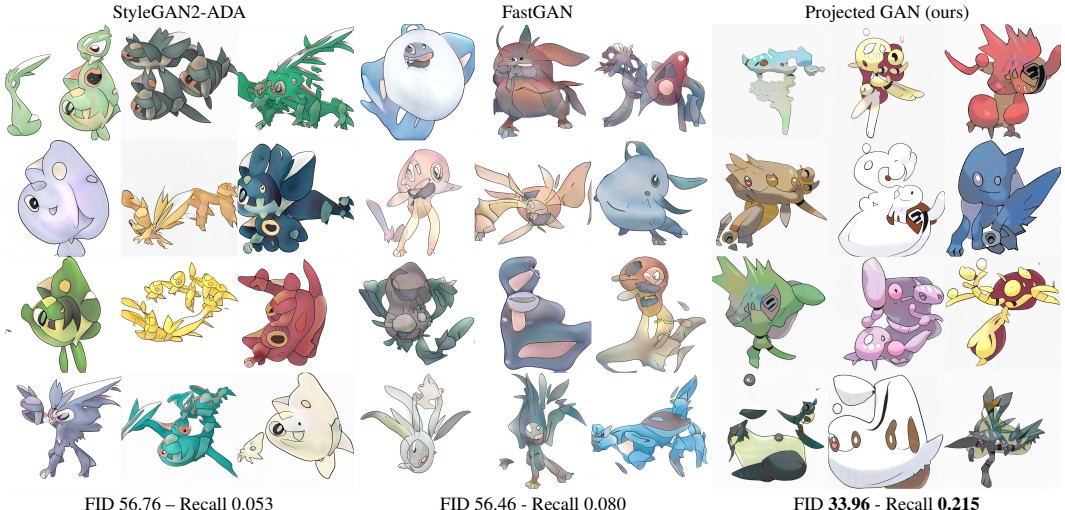

Figure 12: **Uncurated Results for Pokemon** ($1024^2$). The images are selected randomly given one global random seed. We recommend zooming in for comparison.

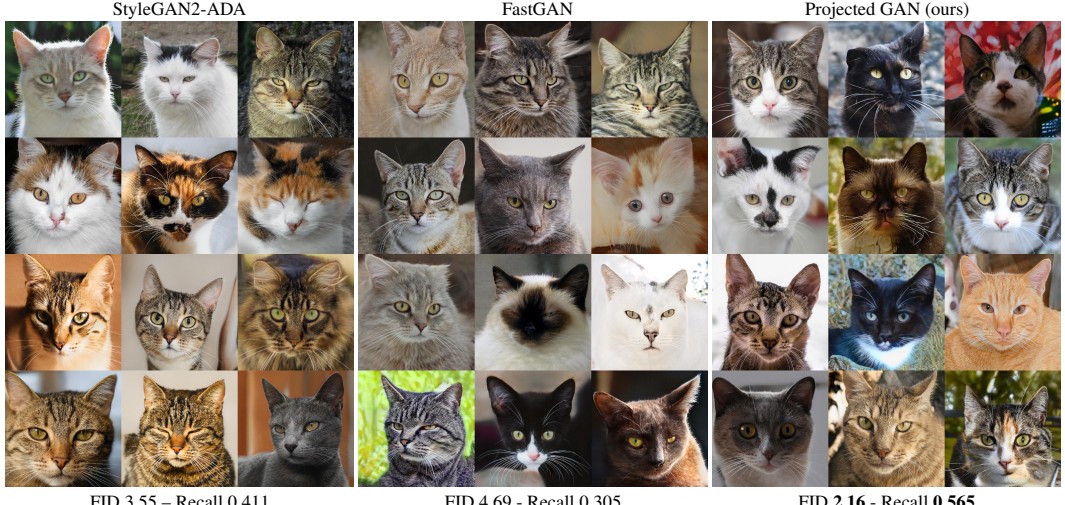

Figure 13: **Uncurated Results for AFHQ-Cat** ($512^2$). The images are selected randomly given one global random seed. We recommend zooming in for comparison.

StyleGAN2-ADA FastGAN Projected GAN (ours)

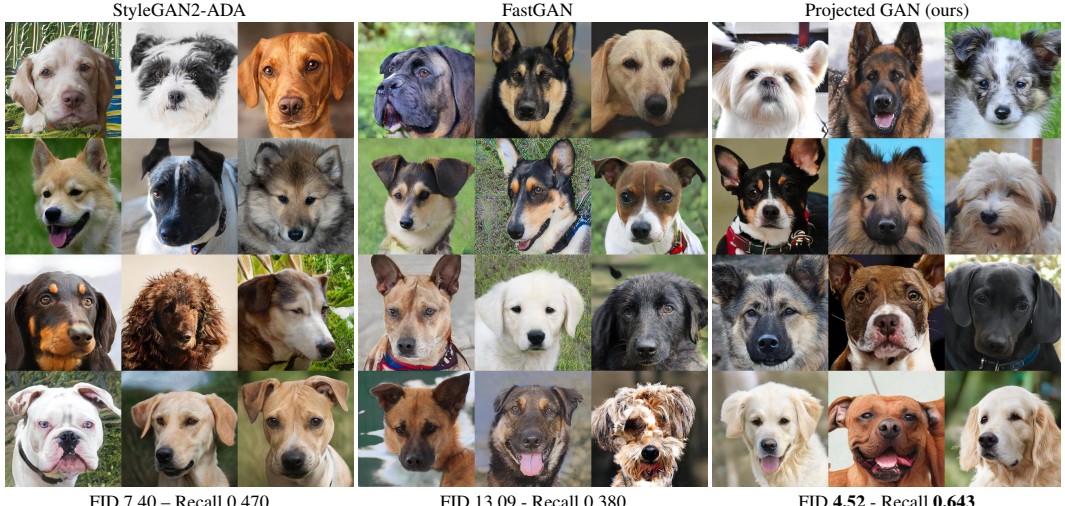

FID 7.40 – Recall 0.470  FID 13.09 - Recall 0.380  FID **4.52** - Recall **0.643**

Figure 14: **Uncurated Results for AFHQ-Dog** ($512^2$). The images are selected randomly given one global random seed. We recommend zooming in for comparison.

StyleGAN2-ADA FastGAN Projected GAN (ours)

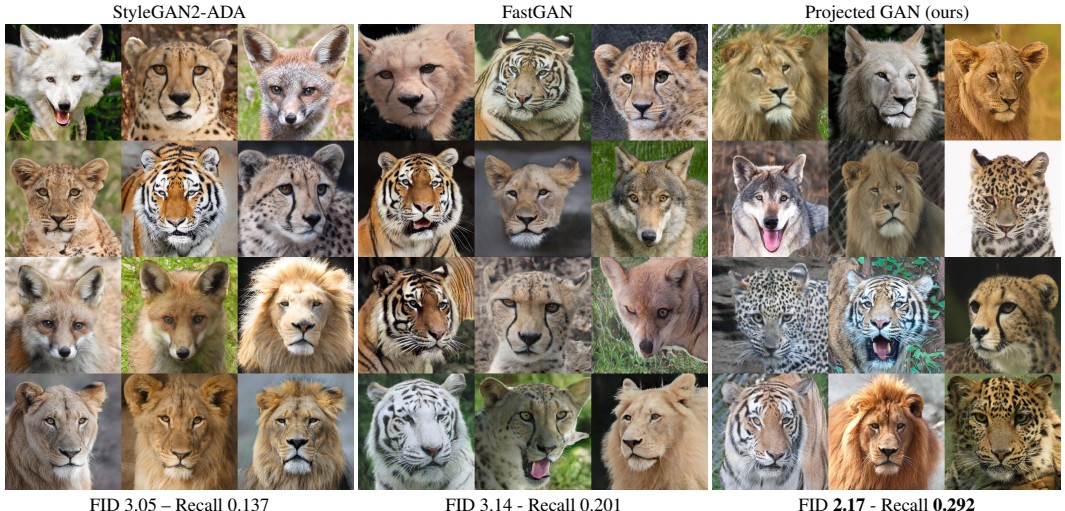

FID 3.05 – Recall 0.137  FID 3.14 - Recall 0.201  FID **2.17** - Recall **0.292**

Figure 15: **Uncurated Results for AFHQ-Wild** ($512^2$). The images are selected randomly given one global random seed. We recommend zooming in for comparison.

# 4 Additional Experiments

This section presents additional experiments referenced in the paper. The experiments explore alternative setups to our final configuration entailing a pretrained, fixed feature network $F$, fixed 1x1 convolutions for cross-channel mixing (CCM), and fixed convolutions for cross-scale mixing (CSM). We follow the setup of Section 4 of the paper: training on LSUN church at a resolution of $256^2$, with a batch size of 64 for 1 million *Imgs*, four discriminators, and a EfficientNet-Lite1 as feature network $F$. Again, we report FID normalized by the FID obtained by a model with a standard single RGB image discriminator. Values $> 1$ indicate worse performance than the RGB baseline. The results are summarized in Table 6.

**No Projection.** In all experiments, we utilize pretrained representations. As a sanity check, it is instructive to test if the architectural bias of $F$ alone is helpful. The results in Table 6 demonstrate that randomly initializing $F$ results in much higher FID.

**CCM.** We explore three different options for CCM. The first option, *Feature Norm*, does not mix features; rather, it normalizes the features to exhibit zero mean and a standard deviation of 1. This option investigates the importance of input scaling. The two other options utilize random convolution with different initializations. *CCM-rotation* is initialized with a random rotation matrix, *CCM-Kaiming* utilizes Kaiming initialization. As shown in Table 6, *CCM-Kaiming* (0.77) improves over the baselines (*Feature Norm* (1.27), *CCM-rotation* (1.27)), over the RGB baseline (1.0), and pretrained $F$ without projection (1.15). This supports our hypothesis, that we need a sufficient amount of channel mixing.

**CSM.** We investigate if training the random projections $P_{rand}$ in CCM and CSM is advantageous for FID. We consider two options: training $P_{rand}$ before or during GAN training. First, we train $P_{rand}$ before GAN training, the feature network $F$ remains fixed. For this purpose, we add a head on the last CSM

| Configuration | FID |
|---|---|
| **No Projection** | |
| Random $F$ | 11.03 |
| Pretrained $F$ | 1.15 |
| **CCM** | |
| Feature Norm | 1.27 |
| CCM-rotation | 1.27 |
| CCM-Kaiming | 0.77 |
| **CCM + CSM** | |
| Denoising AE ($t_0$) | 0.24 |
| Denoising AE ($t_1$) | 0.37 |
| Denoising AE ($t_2$) | 0.44 |
| Denoising AE ($t_3$) | 0.59 |
| Train $F$, Train $P_{rand}$ | 3.09 |
| Fixed $F$, Train $P_{rand}$ | 0.97 |
| Fixed $F$, Fixed $P_{rand}$ | 0.24 |

Table 6: **Ablations.**

layer to map back to full resolution and three output channels, forming an autoencoder architecture. This model trains with a denoising autoencoder loss on ImageNet: the input image is augmented with gaussian blur, JPEG compression, coarse and fine region dropout, and conversion to grayscale. All augmentations are applied with a probability of $0.5$. The target for reconstruction is the non-augmented image. We keep four models, at the beginning of training ($t_0$), at convergence ($t_3$), and two in between with equal spacing in terms of reconstruction loss. After autoencoder training, we use the model (without the additional head) for projected GAN training, denoted as *Denoising AE* ($t_i$). Interestingly, the longer the AE trained, the higher the FID, see Table 6. For the second ablation, different parts of the projection stay fixed during GAN training: (i) training both $F$ and $P_{rand}$, (ii) fixed $F$, training only $P_{rand}$, (iii) both $F$ and $P_{rand}$ stay fixed. Again, the results in Table 6 suggest that training any part of the projection results in worse performance. We conclude that training the random projection is not advantageous, neither before nor during GAN training.

The signed real logits of the discriminator $sign(D(\mathbf{x}))$ are the portion of the training set that gets positive discriminator outputs. Karras et al. [12] find this a helpful heuristic for quantifying discriminator overfitting. The signed logits should remain constant, which they achieve via adapting the augmentation probability during training. Fig. 16 shows that the logits of the RGB baseline steadily increase throughout training, whereas the logits remain mostly constant for Projected GAN training. This observation coincides with the finding that adaptive augmentation is unnecessary for Projected GANs as the logits are already stable.

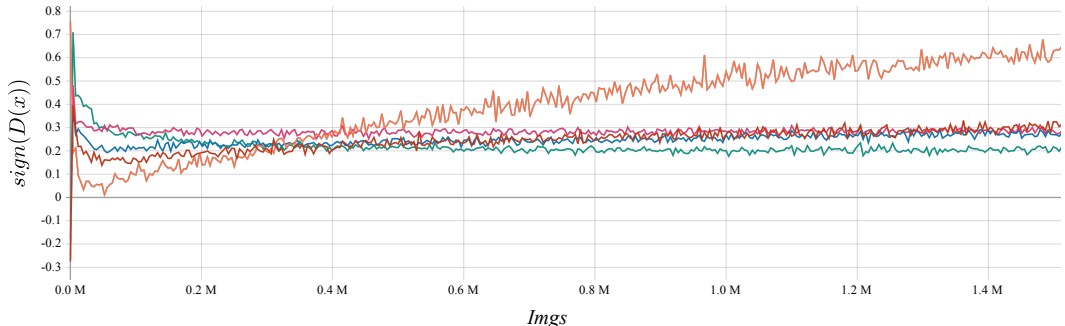

Figure 16: **Signed Discriminator Logits.** For this experiment, we project through $F$ and train with up to four discriminators; we leave the augmentation probability constant. ($|D_i| = 1$: red, $|D_i| = 2$: blue, $|D_i| = 3$: pink, $|D_i| = 4$: green, RGB baseline: orange). For projected GAN training, the logits remain stable throughout training.

# 5 Implementation Details

This section highlights the codebases we used and details hyperparameters and training configurations.

## 5.1 Code and Compute

For dataset preparation, training, and evaluation, we build on top of the Stylegan2-ADA codebase[1]. The evaluation employs the official pretrained Inception network to compute FID and KID. For SwAV-FID, we integrate the model of the SwaV-FID codebase[2] using weights by [3]. SAGAN [32] and Gansformers [8] are trained in the Gansformers codebase[3], we use their default hyperparameters. For the feature networks' implementation and weights, we use timm[4], except for R50-CLIP[5]. Lastly, we utilize official implementations for differentiable augmentation[6] and FastGAN[7].

We conduct our experiments on an internal cluster with several nodes, each with up to 8 Quadro RTX 6000 or NVIDIA V100 using PyTorch 1.7.1 and CUDA 11.0.

## 5.2 Wall Clock-Time

On images at resolution $256^2$, the wall-clock training times measured in sec/kimg using 8 Quadro RTX 6000 are shown in Table 7. StyleGAN2 is the fastest overall, which is expected as we enable mixed-precision and use the custom CUDA kernels provided by the authors. These are not available for the Fast-GAN generator; hence, Projected GAN can only be compared to FastGAN in a fair manner. FastGANs wall-clock times are

| Model | sec/kimg |
|---|---|
| STYLEGAN2-ADA | 5.6 |
| FASTGAN | 7.2 |
| PROJECTED GAN | 10.1 |

Table 7: **Training Speed.**

higher because it uses a reconstruction loss on the discriminator features. This reconstruction loss adds computational overhead. In contrast, projected GANs exhibit lower wall-clock times as we do not need any regularization other than spectral normalization.

## 5.3 Hyperparameters

Below, we describe the hyperparameter search for each method. For optimization, we always use Adam [15] with $\beta_1 = 0, \beta_2 = 0.99$, and $\epsilon = 10^{-8}$. All models employ exponential moving average for the generator weights [11].

**StyleGAN2-ADA.** The official codebase supplies standard configurations of architectures and hyperparameters for different resolutions. Furthermore, an automatic configuration option is available,

---

[1]https://github.com/NVlabs/stylegan2-ada-pytorch

[2]https://github.com/stanis-morozov/self-supervised-gan-eval/

[3]https://github.com/dorarad/gansformer

[4]https://github.com/rwightman/pytorch-image-models

[5]https://github.com/openai/CLIP

[6]https://github.com/mit-han-lab/data-efficient-gans

[7]https://github.com/odegeasslbc/FastGAN-pytorch

entailing several heuristics for different hyperparameters. We find the automatic option very robust for both architecture and most hyperparameters. The exception is the R1 gradient penalty [21], which is highly dependent on the dataset. For all large datasets, the Gansformer codebase suggests suitable values for StyleGAN2. For the small datasets at $256^2$ and $1024^2$, we perform a grid search over $\gamma \in \{1, 10, 20, 50\}$. We first train for 1 M *Imgs*, then continue training only the best one. For all AFHQ datasets, we use the same setting as [12]. All experiments employ adaptive discriminator augmentation with the default target value of $0.6$.

**FastGAN.** We replicate the generator and discriminator architecture of the official FastGAN codebase. FastGAN is robust to most hyperparameters; we always use a learning rate of $0.0002$ and train with a hinge loss. The only sensitive hyperparameter with direct impact on performance is the batch size. Interestingly, FastGAN profits of smaller batch sizes. The default suggested by [20] is a batch size of $8$. We conduct a search over $8, 16, 32$, and $64$, a batch size of $16$ further improves the results, while larger batch sizes decrease performance and even result in divergence in some cases. We employ differentiable augmentation [33] of color, translation, and cutout.

**Projected GAN.** We use the same architecture, learning rate ($0.0002$), batch size ($64$), and hinge loss for all experiments at all resolutions. Compared to FastGAN, we see a slight improvement when increasing model capacity; we double the channel count in each dimension, from a base value of $64$ to $128$. The multipliers for the base value are as follows (resolution: multiplier): $4^2 : 16, 8^2 : 8, 16^2 : 4, 32^2 : 2, 64^2 : 2, 128^2 : 1, 256^2 : 0.5, 512^2 : 0.25, 1024^2 : 0.125$. We did not observe similar improvements for FastGAN when increasing capacity. We employ differentiable augmentation [33] of color, translation, and cutout. Lastly, to extract feature maps of intermediate layers of the feature networks, both CNNs and visual transformers, we follow the protocol presented in the MiDAS [26] codebase[8]. For all EfficientNets and ResNets, we use features at spatial resolutions $r = \{64^2, 32^2, 16^2, 8^2\}$, for DeiT and ViT, we use layers $l = \{3, 6, 9, 12\}$. The CSM blocks follow a residual design typically used in architectures for dense prediction [26]. The lower-resolution feature is passed through a residual 3x3 convolution block; the higher-resolution feature is added, followed by a second residual block and bilinear upsampling, followed by a 1x1 convolution.

The discriminator architectures are shown in Table 8 where $n_i$ are the channels of the different feature network stages, $c_{in}$ and $c_{out}$ are the input and output channels of the DownBlock $DB$. A DownBlock consists of a convolution with k size $k = 4$ and stride $s = 2$, BatchNorm, and LeakyReLU with a slope of $0.2$. We apply spectral normalization on all convolution layers $Conv$.

| Discriminator $L_1$ | Discriminator $L_2$ | Discriminator $L_3$ | Discriminator $L_4$ |
|---|---|---|---|
| $DB(c_i n = c_1, c_o ut = 64)$ | $DB(c_{in} = c_2, c_{out} = 128)$ | $DB(c_{in} = c_3, c_{out} = 256)$ | $DB(c_{in} = c_4, c_{out} = 512)$ |
| $DB(c_{in} = 64, c_{out} = 128)$ | $DB(c_{in} = 128, c_{out} = 256)$ | $DB(c_{in} = 256, c_{out} = 512)$ | $Conv(c_{in} = 512, c_{out} = 1, k = 4)$ |
| $DB(c_{in} = 128, c_{out} = 256)$ | $DB(c_{in} = 256, c_{out} = 512)$ | $Conv(c_{in} = 512, c_{out} = 1, k = 4)$ | |
| $DB(c_{in} = 256, c_{out} = 512)$ | $Conv(c_{in} = 512, c_{out} = 1, k = 4)$ | | |
| $Conv(c_{in} = 512, c_{out} = 1, k = 4)$ | | | |

Table 8: **Discriminator Architectures.**

## Acknowledgments and Disclosure of Funding

We acknowledge the financial support by the BMWi in the project KI Delta Learning (project number 19A19013O). Andreas Geiger was supported by the ERC Starting Grant LEGO-3D (850533). Kashyap Chitta was supported by the German Federal Ministry of Education and Research (BMBF): Tübingen AI Center, FKZ: 01IS18039B and the International Max Planck Research School for Intelligent Systems (IMPRS-IS). Jens Müller received funding by the Heidelberg Collaboratory for Image Processing (HCI). We thank the Center for Information Services and High Performance Computing (ZIH) at Dresden University of Technology for generous allocations of computation time. Lastly, we would like to thank Vanessa Sauer for her general support.

---

[8]https://github.com/intel-isl/MiDaS