# OpenReview forum: "Projected GANs Converge Faster"
_NeurIPS.cc/2021/Conference — NeurIPS 2021 Poster_

### Official Review · Reviewer_CqSv · 2021-07-04

**Rating:** 7
**Confidence:** 4

**Summary:**

This paper proposes to leverage pretrained image feature representations to improve GANs by training the discriminator on random projections of the pretrained features instead of directly on RGB images. This simple substitution results in improved image quality and diversity while also significantly reducing training time compared to existing state-of-the-art methods.

**Limitations And Societal Impact:**

Authors briefly mention benefits of lower computational requirements and the possibility of malicious use such as in deep fakes.

**Main Review:**

Strengths:
- Paper is technically sound. Authors do a good job of first motivating their solution by demonstrating that existing methods cannot leverage deep features from pretrained representations, and then demonstrate that their proposed solution successfully bypasses these limitation.
- Ablation study compares several different SOTA feature representations.
- Generation quality is evaluated on a wide variety of datasets, compared to several other SOTA GAN architectures using a variety of GAN evaluation metrics.
- While the idea of leveraging pretrained feature representations to improve GAN training is certainly not new, the key insight that these representations need to be diluted in order to work well appears to be a novel contribution.
- The reduction in training time afforded by the techniques proposed in this paper could have significant positive impact in several ways, such as: lower barrier to entry for those who would like to train high resolution GANs but don't have the compute budget, faster iteration time for researchers, and less energy consumption/carbon emissions for training models. This is very promising as one of the main challenges with GANs currently is their excessive training time.
- The paper is nicely organized and well written for the most part.

Weaknesses:
- Some clarity issues - see below.

Unfortunately, I had a very difficult time understanding Section 4.1 and Table 1. I was not entirely clear on the experimental setup, particularly in relation to the rows in Table 1, which I could not find descriptions for.
The discriminator architecture could also be described in more detail, even if only in the appendix. Currently the discriminator is described as "a simple convolutional architecture with spectral normalization" (lines 105-106), which is quite ambiguous. Considering that the discriminator is the focus of this paper, I think it is important to include sufficient detail to allow readers to replicate it. Releasing code would certainly help in this regard, but it is nice when a paper is self-contained.

Other questions/comments:
- I found it interesting that most of the performance improvement came in the form of better sample diversity (i.e. recall), rather than just image quality. I think this is a significant result and could be worth mentioning in the main paper rather than having it hidden in the appendix.
- How important is it to use multiple discriminators? i.e., how well would a perceptual discriminator perform if combined with CCM? Or alternatively, how well would FastGAN or StyleGAN2 perform with a separate discriminator for each resolution? (This information may have been in Table 1 but I couldn't tell).
- Considering that it was found that compact representations worked better than high performing classifier representations it would be interesting to test pretrained VGG, especially since it has been so successful for perceptual tasks in the past. (No need to run this experiment now, I am just curious).
- Do you have any intuition as to why FastGAN benefits more from projection compared to StyleGAN2?

**Time Spent Reviewing:**

5

---

> ### Author Response · Authors · 2021-08-10
> **Answer to Reviewer CqSv**
>
> Dear Reviewer CqSv,
>
> Thank you for your comprehensive review and your time. We address your concerns in the following.
>
> __Q1: Section 4.1 and Table1.__
>
> We agree that table 1 is hard to parse, this issue has also been raised by reviewer zAjH. The table means to express what happens when discriminators are applied at different stages of the feature network. To improve clarity, we will add row labels to the table. Moreover, we will relabel the columns with "relative FD" (rel-FD_i) as the values express how much each setting improves over a standard RGB baseline. We hope that this will make the section and table easier to understand.
>
> __Q2: Discriminator Architecture__
>
> The discriminator architectures are as follows:
>
> | Discriminator 1 | Discriminator 2 | Discriminator 3 | Discriminator 4 |
> |---|---|---|---|
> | DownBlock(c_in=c_1, c_out_=64) | DownBlock(c_in=c_2, c_out_=128) | DownBlock(c_in=c_3, c_out_=256) | DownBlock(c_in=c_4, c_out_=512) |
> | DownBlock(c_in=64, c_out_=128) | DownBlock(c_in=128, c_out_=256) | DownBlock(c_in=256, c_out_=512) | Conv2d(c_in=512, c_out=1, kernel=4) |
> | DownBlock(c_in=128, c_out_=256) | DownBlock(c_in=256, c_out_=512) | Conv2d(c_in=512, c_out=1, kernel=4) | |
> | DownBlock(c_in=256, c_out_=512) | Conv2d(c_in=512, c_out=1, kernel=4) | | |
> | Conv2d(c_in=512, c_out=1, kernel=4) | | | |
>
> where n_i are the channels of the different feature network stages, c_in and c_out are the input and output channels of the DownBlock. A single DownBlock consists of a 4x4 Conv, BatchNorm, and LeakyReLU with a slope of 0.2. We apply spectral normalization on all convolution layers. We will add this information to the appendix, thank you for the suggestion.
>
> __Q3: Main improvements are based on improved sample diversity.__
>
> We agree that the improved recall should be mentioned in the main paper and we will add it to section 4. The _recall_ improvements are most notable on large datasets, e.g. church, where the image fidelity appears to be similar to StyleGAN.
>
> __Q4: Importance of multiple discriminators__
>
> We run the experiment perceptual discriminator + CCM/CSM using the same settings as for Table 2. The results are as follows (we report the relative FID_i/FID_rgb) as in the table:
>
> | CCM | rel-FD1 | rel-FD2 | rel-FD3 | rel-FD4 | rel-FID |
> |---|---|---|---|---|---|
> | Discs on {1} | __0.27__ | 0.21 | 0.26 | 0.50 | 0.59 |
> | Discs on {1,2} | __0.27__ | __0.18__ | __0.21__ | __0.41__ | __0.48__ |
> | Discs on {1,2,3} | 0.31 | 0.25 | 0.24 | 0.54 | 0.67 |
> | Discs on {1,2,3,4} | 0.53 | 0.34 | 0.34 | 0.59 | 0.77 |
> | Perceptual Disc | 5.33 | 3.06 | 2.14 | 1.09 | 4.77 |
>
> | CSM | rel-FD1 | rel-FD2 | rel-FD3 | rel-FD4 | rel-FID |
> |---|---|---|---|---|---|
> | Discs on {1} | 0.34 | 0.25 | 0.19 | 0.35 | 0.44 |
> | Discs on {1,2} | __0.21__ | 0.18 | 0.16 | 0.27 | 0.31 |
> | Discs on {1,2,3} | 0.41 | 0.26 | 0.17 | 0.23 | 0.29 |
> | Discs on {1,2,3,4} | 0.26 | __0.16__ | __0.13__ | __0.16__ | __0.24__ |
> | Perceptual Disc | 2.53 | 1.37 | 0.89 | 0.43 | 2.13 |
>
> Accordingly, a perceptual discriminator is inferior to multiple discriminators, with or without random projections.
>
> __Q5: "How well would FastGAN or StyleGAN2 perform with a separate discriminator for each resolution?"__
>
> Regarding this question, we write in section 4.1:
>
> "We also experimented with discriminators at resized versions of the original image, but could not find a setting of hyperparameters and architectures that improves over the single image baseline".
>
> We tried setups similar to the ones with a feature network, i.e., 2-4 discriminators on scales 128x128, 64x64, and 32x32. A closely related idea has been explored in progressive growing GANs [1], but the multi-scale discriminators were discarded in follow-up work. We will add our experiments with discriminators at resized versions of the original image to the supplement.
>
> __Q6: VGG as feature network__
>
> We ran the experiment with VGG11 and VGG16 on church for 2M _Imgs_. The final FIDs are:
>
> | Model | FID |
> |---|:---:|
> | VGG11 | 4.62 |
> | VGG16 | 4.69 |
> | EfficientNet-Lite1 | 3.02 |
>
> Both VGGs achieve reasonably low FID but perform worse than our standard feature network. Interestingly, the VGGs do not differ much in their results, which is likely related to both configurations having the same number of channels per feature stage (128, 256, 512, 512).
>
> __Q7: Why FastGAN benefits more from projection__
>
> As shown in Figure 5, samples for fixed noise undergo significant changes during training. The architecture of a FastGAN generator, a simple image pyramid with skip connections, appears to keep up better with the semantic feedback passed through the feature network. StyleGAN2 is inherently more constrained, i.e., the mapping network and style modulation lead to slower adaptation of the generator. In our experiments, we observe that the discriminator can quickly overpower the generator for suboptimal learning rates. Finding faster StyleGAN-like architectures might be a solution to this problem which will be interesting to explore in future work.
>
> Please let us know if you still have any remaining doubts or questions. We are eager to enhance our manuscript further and possibly improve upon your initial evaluation score.
>
> __References__
>
> [1] Karras et al. "Progressive growing of gans for improved quality, stability, and variation.", ICLR, 2018.

---

> ### Author Response · Authors · 2021-09-10
> **More Questions?**
>
> Dear Reviewer CqSv,
>
> Thank you again for your review. We hope that our rebuttal could address all your questions and concerns regarding unclear sections in the initial draft, missing details about the discriminator architecture, and additional ablations of the discriminator configuration. As the discussion phase is nearing its end, we wondered if you might still have any concerns that we could address.
>
> Thank you for your time.

---

### Official Review · Reviewer_UArS · 2021-07-07

**Rating:** 5
**Confidence:** 4

**Summary:**

The papers proposes a technique for faster training of GANs while achieving state-of-the-art FID scores on several datasets. The main idea is to use pre-trained features for the discriminator. Four improvements are recommended: 1) Multi-scale discriminators, 2) cross-channel mixing 3) cross-scale mixing 4) choice of pre-trained network.

FID scores are shown for several datasets, CLEVR, FFHQ, Cityscapes, Bedroom and Church.

**Limitations And Societal Impact:**

The paper did a great job of calling out their limitations in Section 6. No societal impact was provided beyond line 326.

**Main Review:**

The paper presents very impressive FID results across numerous datasets. However, it is unclear why these improved results were achieved. It is possible the paper could be accepted if the results could be clarified, but in its current state it is below the bar for acceptance.

Positives:

\++ Very good FID scores on many datasets.

\+ Very fast to train when compared to other approaches.

\+ Good job in calling out the approach's limitations.

Negatives:

\- It is unclear why CCM would make any difference in the network. A linear transformation before a linear layer is equivalent to a single linear layer. Why would CCM help training? This is particularly confusing since the labels in Table 1a are missing.

\- For CSM, did the authors compare against a discriminator that has access to all feature layers? It is unclear if this would help if the discriminators weren't independent.

\- The y-axis labels are missing in Table 1a and 1b. Without these labels it is hard to determine what the numbers actually mean.

\- Why are the values in Table 2 different from Table1? I.e. the FID scores are >1 in Table 2 but below 0.5 in Table 1. Is it because training was done for 10M images instead of 1M? If so, why are the results worse for larger training set sizes?

\- Is the reason the FID scores are so high due to the discriminator features being essentially classifier features? If so, the use of FID scores is suspect for evaluation. To be convincing, it would be better to run human studies or another metric that indicates the generated images are actually better. It is not clear from the supplementary material that the images generated are actually improved.



**Time Spent Reviewing:**

2.5 hours

---

> ### Author Response · Authors · 2021-08-10
> **Answer to Reviewer UArS**
>
> Dear Reviewer UArS,
>
> Thank you for your review and your time. We will address your concerns in the following.
>
> __Q1: Why would CCM help training?__
>
> As soon as we start training the CCM layer, your intuition is correct: the final results are the same with or without CCM as shown in Appendix 4. While it is true that CCM is a single linear layer, it can still strongly modify how information is provided to the downstream discriminator as its weights stay fixed. For example, a 1x1 convolution with a single output channel loses a lot of information. More output channels lead to less or no information loss, as we described, but the discriminator is still never provided with the original features. We also clarify the label issue of Table 1 below in Q4.
>
> __Q2: CSM + Perceptual Discriminator:__
>
> As we understand, your question aims at combining a perceptual discriminator, which has access to all feature layers, with random projections. For this rebuttal, we ran an experiment with perceptual discriminator + CCM/CSM using the same settings as for Table 2. The results are as follows (we report the relative FID, i.e.,  FID_i/FID_rgb):
>
> | CCM | rel-FD1 | rel-FD2 | rel-FD3 | rel-FD4 | rel-FID |
> |---|---|---|---|---|---|
> | Discs on {1} | __0.27__ | 0.21 | 0.26 | 0.50 | 0.59 |
> | Discs on {1,2} | __0.27__ | __0.18__ | __0.21__ | __0.41__ | __0.48__ |
> | Discs on {1,2,3} | 0.31 | 0.25 | 0.24 | 0.54 | 0.67 |
> | Discs on {1,2,3,4} | 0.53 | 0.34 | 0.34 | 0.59 | 0.77 |
> | Perceptual Disc | 5.33 | 3.06 | 2.14 | 1.09 | 4.77 |
>
> | CSM | rel-FD1 | rel-FD2 | rel-FD3 | rel-FD4 | rel-FID |
> |---|---|---|---|---|---|
> | Discs on {1} | 0.34 | 0.25 | 0.19 | 0.35 | 0.44 |
> | Discs on {1,2} | __0.21__ | 0.18 | 0.16 | 0.27 | 0.31 |
> | Discs on {1,2,3} | 0.41 | 0.26 | 0.17 | 0.23 | 0.29 |
> | Discs on {1,2,3,4} | 0.26 | __0.16__ | __0.13__ | __0.16__ | __0.24__ |
> | Perceptual Disc | 2.53 | 1.37 | 0.89 | 0.43 | 2.13 |
>
> Accordingly, a perceptual discriminator is inferior to multiple discriminators, with or without random projections. Therefore, your proposition is correct; CSM needs to be used with independent discriminators to be effective. We will add these results to the supplementary.
>
> __Q3: Missing Labels in Table 1__
>
> We will add row labels to Table 1 as in the table above (Discs on {1,...}). Also, we will relabel the columns with "relative FD" (rel-FD_i) as the values express how much the setting improved over a standard RGB baseline. We hope that this will make the section and table easier to understand.
>
> __Q4: Why are values in Table 2 different from Table 1__
>
> The values in table 1 refer to the relative FID, i.e., how much the setting improved over a standard RGB baseline. We hope that relabelling the columns in Table 1 will improve clarity.
>
> __Q5: Correlation with FID scores__
>
> We followed your suggestion and conducted a human preference study with 28 participants unfamiliar with our method for this rebuttal. The structure of the study is as follows:
>
> - For each dataset, we first show samples of the real dataset for context.
> - We then present two sample sheets and ask the participants to rank the sheets relative to each other based on sample fidelity and diversity. We define these as follows: (i) Fidelity is the degree to which the generated samples resemble the real ones. (ii) Diversity measures whether the generated samples cover the full variability of the real samples.
> - Each sheet contains nine random samples; we present three sample sheet pairs per dataset.
> - Samples and comparison pairs are randomized per participant. We make sure that all possible pairings are equally represented.
> - Evaluated Models: StyleGAN2-ADA, FastGAN, Projected GAN, and real data (for control)
> - Evaluated datasets: all 256x256 datasets
> - We count how often a given model wins a comparison and report the relative amount of wins.
>
>  The results are as follows:
>
> | _Fidelity_ | CLEVR | FFHQ | Cityscapes | Bedroom | Church | ArtPainting | Landscape | AnimalFace | Flowers | Pokemon | All Datasets |
> |:---:|:---:|:---:|:---:|:---:|:---:|:---:|:---:|:---:|:---:|:---:|:---:|
> | Stylegan2-ada | 14 % | 32 % | 17 % | 5 % | 16 % | 16 % | 21 % | 4 % | 17 % | 10 % | 15 % |
> | FastGAN | 7 % | 2 % | 15 % | 3 % | 9 % | 0 % | 8 % | 7 % | 4 % | 0 % | 6 % |
> | Projected GAN | 42 % | 5 % | 15 % | 16 % | 9 % | 28 % | 17 % | 55 % | 25 % | 38 % | 25 % |
> | Data | 37 % | 61 % | 53 % |  76 % | 66 % | 56 % | 54 % | 34 % | 54 % | 52 % | 54 % |
>
> | _Diversity_ | CLEVR | FFHQ | Cityscapes | Bedroom | Church | ArtPainting | Landscape | AnimalFace | Flowers | Pokemon | All Datasets |
> |:---:|:---:|:---:|:---:|:---:|:---:|:---:|:---:|:---:|:---:|:---:|:---:|
> | Stylegan2-ada | 13 % | 24 % | 9 % | 10 % | 19 % | 11 % | 25 % | 9 % | 17 % | 6 % | 14 % |
> | FastGAN | 13 % | 4 % | 11 % | 2 % | 0 % | 4 % | 13 % | 9 % | 17 % | 0 % | 7 % |
> | Projected GAN | 31 % | 12 % | 21 % | 21 % | 14 % | 27 % | 16 % | 49 % | 29 % | 40 % | 27 % |
> | Data | 43 % | 60 % | 59 % | 67 % | 67 % | 58 % | 46 % | 33 % | 37 % | 54 % | 52 % |
>
> The ranking above largely agrees with the results obtained via FID. On FFHQ, the study demonstrates our reported failure case for projected GANs. Interestingly, on AnimalFace projected GAN outperforms real data. We hypothesize that this is due to the fact that for AnimalFace there is a significant portion of low-quality images (blurry, compression artifacts) in the dataset, and possibly projected GAN generates fewer of those. Of course, human studies are not optimal, as it is not straightforward to evaluate sample diversity - which is a strong suit of our method - given only a few samples.
>
> Other reviewers have also raised the issue of a possible correlation between metric and training objective. Therefore, we also evaluated non-ImageNet FIDs, see our response to reviewer gvfM, and the non-neural sliced Wasserstein distance, see our response to reviewer zAjH.
>
> We hope our response could address all of your questions and that you might consider raising your initial score. Please let us know if you still have any remaining doubts or questions.

---

> ### Author Response · Authors · 2021-09-10
> **More Questions?**
>
> Dear Reviewer UArS,
>
> Thank you again for your review. We hope that our rebuttal could address all your questions and concerns regarding the used evaluation metric, the effectiveness of CCM, and ablations of the perceptual discriminator. As the discussion phase is nearing its end, we wondered if you might still have any concerns that we could address.
>
> Thank you for your time.

---

### Official Review · Reviewer_zAjH · 2021-07-16

**Rating:** 7
**Confidence:** 4

**Summary:**

This paper investigates the use of pre-trained imagenet classifier networks as feature extractors for GAN discriminators. The author propose to use multiscale fixed random projections combined via an FPN-like approach, and explore which levels of the feature hierarchy are best fed into an ensemble of discriminators under different projection and mixing settings. Results are presented on a range of unimodal datasets (CLEVR, FFHQ, LSUN, etc) and compared against FastGAN and StyleGAN baselines, showing improvements particularly wrt sample efficiency.

**Limitations And Societal Impact:**

yes

**Main Review:**

My take:
This is a decent paper. For the most part, it is clear (with a few sharp edges, noted in my detailed notes) and easy to follow. The experiments are sensible and the results are reasonably strong, particularly with respect to sample efficiency.

My primary concern with this paper would be that using pretrained features from classifiers in any way tends to strongly improve performance on *any metric which involves a pretrained classifier net,* (even one trained in a self-supervised fashion), but does not necessarily lead to actual sample quality improvements. E.g., for ImageNet GANs, this type of supervision leads to generating images which are more centred and emphasize the foreground, strongly improving both the rate of observed FID reduction and the final performance, but direct visual comparison makes it quite clear that the overall sample quality is not actually changed meaningfully.

While I appreciate that the authors include multiple metrics aside from FID in the appendix, these metrics are still nonetheless based on pretrained features which are susceptible to this phenomenon. I think this manuscript would be made substantially more airtight if non-pretrained-net-based metrics were used: reporting LPIPS and the sliced Wasserstein distance would go a long way on this front. Normally I wouldn’t much care to see these metrics, but because of this specific concern I think it pertinent. Based on the presented samples I do not expect the relative ordering of the comparisons to change.  In particular it is clear that regardless of final performance, the sample complexity is sharply improved by this method, so this is to me more a matter of adhering to standards of scientific rigour by ensuring that one cannot explain away the improvements by appealing to the aforementioned confounding factor.

Ultimately I think this paper will be of interest to practitioners operating in low-resource (data or compute settings), but not of especial interest to the field of (deep) generative modeling at large. That’s not a negative per se, but it does bear thinning about when guessing at its potential impact at NeurIPS. I’m rating this paper around a 6.5, and I expect a healthy discussion phase; if the authors address my concerns wrt metrics I would likely be willing to move up to a solid 7.


Notes:

-Section 3 acknowledges that if the feature extractor tosses information then the generative modeling objective is changed; it would be good to devote slightly more discussion to this on a theoretical level, perhaps in future work. I appreciate the discussion in Section 6 which demonstrates artifacts and potential negative downsides of the approach in this context.

“We observe that features at deeper layers are significantly harder to fit, as evidenced by our experiments in Section 4. We hypothesize that a discriminator can focus on a few descriptive feature channels while wholly disregarding others.”

This couplet needs a rework for several reasons. First, it seems the authors don’t mean that these features are harder to fit, but harder to *cover* (in the reverse KL sense); it sounds like these features are easier to overfit to. Second, this hypothesis is odd: focusing on a subset of the channels does not necessarily mean that D is overfitting, as it’s not only possible but highly likely that the feature space of any given pretrained network is not wholly utilized, so focusing just on a few channels may simply be because those channels are indeed the most informative. I think I understand the point the authors are trying to make, but they need to be careful and very specific with language here. Perhaps “focus on a subset of the feature space” would be a better choice

-” We do not choose an inception network [59] to avoid strong correlations with the evaluation metric FID”
While I understand the authors motivations here, this is misleading--using *any* classifier network will lead to strong correlations with the FID metric, regardless of if it is the same inception network used for the metric or not. This reviewer speaks directly from empirical experience, but evidence in the literature (that most classifier nets tend to learn approximately the same thing when trained on a given dataset) abounds. The supplementary provides several other metrics which should ameliorate this to some degree, but as mentioned in my main review the core issue is the use of classifier features to evaluate a model fit to classifier features, not the particular distance/divergence/Precision-recall curve used to evaluate the discrepancy between said features.


-In Table 1, the rows are unlabeled, and even after spending quite a bit of time reading the text, the caption, and examining the table, I was unable to ascertain with certainty exactly what each row corresponds to. Please label the rows of table 1 or explicitly explain in clear and direct language what they represent. I *think* the table is only meant to show what happens when you apply discriminators to different feature layers (with the highlights indicating which of the seven stages in an EffNet are being used as features), but this is not explained and doesn’t explain the meaning of the different values as one goes down a column. The column labels for this table are also problematic: they’re labeled as “FD” but they actually represent the ratio of performance relative to an RGB only baseline; while this is explained in the text, this is impossible to infer from the table itself and further adds to the confusion.


-The reduction in FD performance as discriminators are added to later layers is indeed interesting, but could potentially also be explained away by the known fact that (without contrary intervention) classifier networks tend to heavily rely on local structure and texture rather than global structure, and this is emphasized even in their deeper layers. While non-FID metrics are included in the appendix for the final models, it would be good to have them for this table as well.

-The authors mention wallclock time in several places but do not provide direct comparisons in terms of per-step training latency. These should be measured and included: since the sample complexity of the proposed method appears to outstrip the baselines by a large margin, then even if the use of the pretrained model were to incur a modest relative compute overhead, this would still favor the authors. In other words, what is the value of training imgs/s on a given GPU for each of the different methods?  A small table in the supplementary showing this would suffice and shouldn’t require any additional experiments, just the authors checking their training logs.


**Time Spent Reviewing:**

5

---

> ### Author Response · Authors · 2021-08-10
> **Answer to Reviewer zAjH**
>
> Dear Reviewer zAjH,
>
> Thank you for your comprehensive review and your time. We address your concerns in the following, starting with the primary one.
>
> __Q1: Correlation with FID scores__
>
> Other reviewers also share your concern about a possible correlation between our training objective and FID, and we agree on its significance. Including a non-neural metric, i.e., sliced Wasserstein distance (SWD), is a great idea; thank you for your suggestion. We follow the protocol proposed by [1] to compute SWD and obtain the following results for the 256x256 datasets (we report the mean over all SWD resolutions):
>
> | _SWD x 10e-3_ | CLEVR | FFHQ | Cityscapes | Bedroom | Church | ArtPainting | Landscape | AnimalFace | Flowers | Pokemon |
> |:---:|:---:|:---:|:---:|:---:|:---:|:---:|:---:|:---:|:---:|:---:|
> | Stylegan2-ada | 20.82 | 10.38 | 10.71 | 12.53 | 14.62 | 25.55 | 19.06 | 22.31 | 14.04 | 14.73 |
> | FastGAN | 28.51 | 10.19 | 9.45 | 14.68 | 14.42 | 21.94 | 29.87 | 29.23 | 17.39 | 46.81 |
> | Projected GAN | __12.90__ | __6.41__ | __7.27__ | __6.83__ | __8.37__ | __11.44__ | __15.38__ | __14.34__ | __9.61__ | __11.65__ |
>
> The ranking mirrors the ones of FID and SwaV-FID. However, SWD might exhibit different, unknown shortcomings. For example, according to SWD, there is no difference between Stylegan2-ADA and FastGAN on LSUN church, even though the StyleGAN2 samples are clearly superior. These results suggest that a new non-neural metric might be helpful for projected GANs. Apart from SWD, we also evaluate non-ImageNet FIDs, see our response to reviewer gvfM, and a human preference study, see our response to reviewer UArS.
>
> In this context, we like to enquire about your following statement:
>
> "[...] using pretrained features [...] tends to strongly improve performance on any metric which involves a pretrained classifier net[...], e.g., for ImageNet GANs, this type of supervision leads to generating images which are more centred and emphasize the foreground, strongly improving both the rate of observed FID reduction and the final performance, but direct visual comparison makes it quite clear that the overall sample quality is not actually changed meaningfully."
>
> Are you aware of any related GAN literature supporting this claim? The claim is likely valid and backed by our failure cases on AFHQ, but we are unaware of any published demonstration of that issue. We would be very interested in pointers to relevant work and, of course, happy to cite and discuss them. Furthermore, while the statement is presumably sensible when using classifier features for projected GAN training, it is not clear if a strong emphasis on a central object would still occur for self-supervised (eg, contrastive) features or CLIP features.
>
> __Q2: Possible impact to the field of deep generative modeling__
>
> We respectfully disagree with the assessment of the possible impact of our work:
>
>  "Ultimately I think this paper will be of interest to practitioners operating in low-resource (data or compute settings), but not of especial interest to the field of (deep) generative modeling at large"
>
> GANs are a significant research branch of generative modeling. Hence, a considerable change in the standard GAN training paradigm will be of interest to the field at large. As also noted by Reviewer CqSv, one of the major challenges of GANs is their excessive training time, requiring a high computational budget and leading to high energy consumption. A second big challenge of GANs is training stability and mode collapse. Projected GANs significantly improve both training stability and training time. We like to reemphasize that projected GANs improve the final results on large datasets (Church (130k images), CLEVR (70k images), and bedroom (1.3 M images)) across all metrics (neural and non-neural), especially also in terms of sample diversity as demonstrated by the recall metric reported in our supplementary and by the human preference study conducted in response to UArS.
>
> __Q3: Reformulation: "We observe that features at deeper layers are significantly harder to fit…"__
>
> Thanks for the suggestion. We agree. We will reformulate the passage to: "We observe that features at deeper layers are significantly harder to cover, as evidenced by our experiments in Section 4. We hypothesize that a discriminator can focus on a subset of the feature space while wholly disregarding other parts."
>
> __Q4: "We do not choose an inception network to avoid strong correlations"__
>
> We agree with your points as also discussed in our response to Q1. We will include and refer to the newly evaluated non-classifier and non-neural metrics in the revised version of the paper.
>
> __Q5: Labelling of Table 1__
>
> Your interpretation of table 1 is correct: "I think the table is only meant to show what happens when you apply discriminators to different feature layers (with the highlights indicating which of the seven stages in an EffNet are being used as features)"
>
> We understand that the missing row labels and the "FD" column labels add confusion. We will add row labels, relabel the columns with "relative FD" (rel-FD_i), and add a better explanation to the caption. We hope that these changes will improve clarity.
>
> __Q6: FD reduction as discriminators are added__
>
> Your hypothesis that the reduction in FD" could [...] be explained away by the known fact that [...] classifier networks tend to heavily rely on local structure" is indeed interesting; we will mention it in the text.
>
> __Q8: Wall Clock-Time__
>
> On 256x256 the wall-clock training times measured in sec/kimg using 8 Quadro RTX 6000 are as follows:
>
> | Model | sec/kimg |
> |---|:---:|
> | StyleGAN2-ADA | 5.6 |
> | Projected-GAN | 7.2 |
> | FastGAN | 10.1 |
>
> StyleGAN2 is the fastest overall, which is expected as we enable mixed-precision and use the custom CUDA kernels provided by the authors [2]. These are not available for the FastGAN generator; hence, Projected GAN can only be compared to FastGAN in a fair manner. FastGANs wall-clock times are higher because it uses a reconstruction loss on the discriminator features for regularization. This reconstruction loss adds computational overhead. In contrast, projected GANs exhibit lower wall-clock times as we do not need any regularization other than spectral normalization. We will add this comparison to the appendix.
>
> Thank you again for your extensive review. We hope that our arguments are convincing and that you might consider raising your initial score to a 7 or possibly an 8. Please let us know if you still have any remaining doubts or questions which we are happy to address.
>
> __References__
>
> [1] Karras et al. "Progressive growing of gans for improved quality, stability, and variation.", ICLR, 2018.
>
> [2] Karras, et al. "Training generative adversarial networks with limited data.", NeurIPS, 2020

---

> > ### Comment · Reviewer_zAjH · 2021-09-10
> > **Response**
> >
> > Hi Authors,
> >
> > My apologies for my late response (trying times personally). I appreciate the author's rebuttal and feel that (especially regarding metrics) it has satisfactorily addressed my concerns; I hope the authors feel that adding the additional metrics will strengthen their paper. I still hold to my claim on the expected impact but as predicting impact is of course as fraught as predicting bitcoin prices, I don't see this as an issue. I increase my score to a solid 7 and support acceptance.

---

> > > ### Author Response · Authors · 2021-09-10
> > > **Answer**
> > >
> > > Dear Reviewer zAjH,
> > >
> > > Thank you for your answer; we are happy that we could address your concerns. Thank you again for your constructive feedback; we indeed think that it helped to improve our paper.
> > >
> > > Regarding your note, you are correct; it is supposed to be sec/kimg, and refers to training time; we edited the rebuttal answer. While we agree that imgs/sec also makes sense, we would like to stick to sec/kimg as this is what the popular official code base of StyleGAN2-ADA uses (reports sec/kimg for various GPU setups, uses it for logging, etc.).

---

> > ### Comment · Reviewer_zAjH · 2021-09-10
> > **One note**
> >
> > I would also note however that the authors wall clock time table in their rebuttal is either mislabeled or their interpretation of it is factually incorrect: if the y axis is kimg/s then FastGAN is the fastest model. Using seconds/kimg seems odd to me so I would recommend keeping it in terms of imgs/s (where higher is better).
> >
> >
> > Additionally, please ensure it is clear if this is training wall clock time or inference wall clock time. Based on the comment regarding reconstruction losses I assume it is training time, but this should be explicitly stated.

---

> ### Author Response · Authors · 2021-09-10
> **More Questions?**
>
> Dear Reviewer zAjH,
>
> Thank you again for your review. We hope that our rebuttal could address all your questions and concerns regarding the used evaluation metric and unclear formulations in the initial draft. As the discussion phase is nearing its end, we wondered if you might still have any concerns that we could address.
>
> Thank you for your time.

---

### Official Review · Reviewer_gvfM · 2021-07-16

**Rating:** 7
**Confidence:** 4

**Summary:**

The demonstrate that effectiveness of the results on a wide variety of datasets and outperform current SOTA on the several benchmarks. They show that they get faster convergence than competing works like StyleGAN2 and show excellent qualitative results on very small datasets like the provided pokemon.

**Limitations And Societal Impact:**

Yes.

**Main Review:**

# Originality:
I am not aware of any GANs that use pre-trained ImageNet embeddings in a GAN like so. Projects real and generated samples into a fix pre-trained feature space. There are some work that has used pretrained imagenet features to create knn clusters that are later used an auxilary classification task for the discriminator, but this approach seems sufficiently different to those approaches. The authors

# Quality:
 Is the submission technically sound? Are claims well supported (e.g., by theoretical analysis or experimental results)? Are the methods used appropriate? Is this a complete piece of work or work in progress? Are the authors careful and honest about evaluating both the strengths and weaknesses of their work?

Most of the claims in the paper are technically sound. The one issue I have is the entanglement between using ImageNet feature and entanglement with FID score. It's true they use a different architecture, EfficientNet than the InceptionNet does help a bit in that regard as stated in the paper. "We do not choose an inception network [59] to avoid strong150correlations with the evaluation metric FID." Inherently, EfficientNet is trained on the same data as InceptionNet though, so I would prefer if a non-ImageNet trained feature space was used for evaluation. Even another similar dataset that could be use for classification like Google OpenImages, MS CoCo classification, etc. That would strengthen the results in the paper. I do appreciate that they include other metrics like SwAV-FID (although this should be moved to the main paper as it's the most compelling quantitative result.

Given the use of self-supervised FID assuages my concerns though. Also the Pokemon dataset results are pretty compelling. I'd also like to point that ImageNet and StyleGAN's architecture have similarly object centric biases (the objects are always in the center of the frame even in the Pokemon example).

# Clarity:
The submission is fairly well written and the qualitative examples are very convincing. The paper is very well written and the supplement is detailed.


# Significance:
The paper proposes a pre-trained feature layer to accelerate GAN training and improve results on small datasets. They show impressive results and demonstrates which techniques work and some that don't on guiding the GAN's using the feature spaces. It also shows how to utilize feature space like ImageNet on categories that are not included within. It also demonstrates some of the limitations of the approach like that fact that it actually hampers the results on certain datasets like FFHQ.



**Time Spent Reviewing:**

3

---

> ### Author Response · Authors · 2021-08-10
> **Answer to Reviewer gvfM**
>
> Dear Reviewer gvfM,
>
> Thank you for your review and your time. We understand that your primary concern is the possible correlation between the evaluation metric FID and the feature network. As you mentioned, we include the evaluation of SwaV-FID in the appendix, which reflects the ranking obtained via FID, and we will highlight these results in the main paper.
>
> The SwaV-ResNet uses a different objective. However, we agree that a possible confounding factor might be the training data as SwaV is also trained on ImageNet. Thus, we follow your suggestion of using a metric with non-ImageNet features. For this purpose, we evaluate CLIP-FID (using a ResNet50 trained with the CLIP objective on the dataset collected by [1]) and VirTex-FID (using a ResNet50 trained on COCO Captions with the VirTex objective [2]). We obtain the following results for the 256x256 datasets:
>
> | _CLIP-FID_ | CLEVR | FFHQ | Cityscapes | Bedroom | Church | ArtPainting | Landscape | AnimalFace | Flowers | Pokemon |
> |:---:|:---:|:---:|:---:|:---:|:---:|:---:|:---:|:---:|:---:|:---:|
> | Stylegan2-ada | 4.10 | 16.50 | 5.88 | 42.12 | 15.85 | 44.13 | 24.89 | 46.18 | 26.30 | 13.96 |
> | FastGAN | 4.24 | 19.23 | 6.46 | 31.10 | 35.47 | 40.47 | 19.84 | 54.69 | 40.12 | 87.65 |
> | Projected GAN | __0.80__ | __7.55__ | __2.96__ | __11.97__ | __13.71__ | __22.91__ | __13.71__ | __16.89__ | __15.83__ | __9.93__ |
>
> | _VirTex-FID_ | CLEVR | FFHQ | Cityscapes | Bedroom | Church | ArtPainting | Landscape | AnimalFace | Flowers | Pokemon |
> |:---:|:---:|:---:|:---:|:---:|:---:|:---:|:---:|:---:|:---:|:---:|
> | Stylegan2-ada | 0.88 | 2.20 | 1.15 | 2.20 | 1.10 | 4.15 | 2.78 | 8.83 | 3.25 | 3.69 |
> | FastGAN | 0.64 | 2.47 | 1.48 | 2.66 | 3.61 | 5.72 | 3.86 | 9.41 | 4.08 | 17.49 |
> | Projected GAN | __0.35__ | __0.64__ | __0.49__ | __0.81__ | __0.82__ | __3.53__ | __1.98__ | __3.79__ | __2.19__ | __2.55__ |
>
> The rankings obtained via CLIP-FID and VirTex-FID mirror the ones of FID and SwaV-FID. To reduce doubts further, we have now also evaluated two non-neural metrics: Sliced Wasserstein Distance (SWD), see our response to reviewer zAjH, and a human preference study, see our response to reviewer UArS.
>
> We hope that these additional results could adequately address your concerns. Please let us know if you still have any remaining doubts or questions. We are happy to enhance our manuscript further and possibly improve upon your initial evaluation score.
>
> __References__
>
> [1] Radford, et al. "Learning transferable visual models from natural language supervision." arXiv, 2021.
>
> [2] Desai and Johnson. "Virtex: Learning visual representations from textual annotations." CVPR, 2021.

---

> ### Author Response · Authors · 2021-09-10
> **More Questions?**
>
> Dear Reviewer gvfM,
>
> Thank you again for your review. We hope that our rebuttal could address all your questions and concerns regarding the used evaluation metric. As the discussion phase is nearing its end, we wondered if you might still have any concerns that we could address.
>
> Thank you for your time.

---

### Decision · Program_Chairs · 2021-09-27

**Decision:**

Accept (Poster)

**Comment:**

The submission discusses projected GANs, i.e., GANs which project generated and real samples into a fixed, pretrained feature space. The initial reviewer assessment is generally positive. The reviewers provided further clarifications in the rebuttal which clarified questions. Reviewers remained assured about their initial assessment. AC thinks that this is a valuable study that will be of interest.